# C-NAV: Towards Self-Evolving Continual Object Navigation in Open World

**Ming-Ming Yu**[1], **Fei Zhu**[2†], **Wenzhuo Liu**[3,4], **Yirong Yang**[1],
**Qunbo Wang**[6*], **Wenjun Wu**[1,5], **Jing Liu**[3,4]

[1]Beihang University, [2]Centre for Artificial Intelligence and Robotics, HKISI-CAS
[3]Institute of Automation, Chinese Academy of Sciences
[4]University of Chinese Academy of Sciences
[5]Hangzhou International Innovation Institute, Beihang University, [6]Beijing Jiaotong University
mingmingyu@buaa.edu.cn    wangqb6@outlook.com

## Abstract

Embodied agents are expected to perform object navigation in dynamic, open-world environments. However, existing approaches typically rely on static trajectories and a fixed set of object categories during training, overlooking the real-world requirement for continual adaptation to evolving scenarios. To facilitate related studies, we introduce the continual object navigation benchmark, which requires agents to acquire navigation skills for new object categories while avoiding catastrophic forgetting of previously learned knowledge. To tackle this challenge, we propose C-Nav, a continual visual navigation framework that integrates two key innovations: (1) A dual-path anti-forgetting mechanism, which comprises feature distillation that aligns multi-modal inputs into a consistent representation space to ensure representation consistency, and feature replay that retains temporal features within the action decoder to ensure policy consistency. (2) An adaptive sampling strategy that selects diverse and informative experiences, thereby reducing redundancy and minimizing memory overhead. Extensive experiments across multiple model architectures demonstrate that C-Nav consistently outperforms existing approaches, achieving superior performance even compared to baselines with full trajectory retention, while significantly lowering memory requirements. The code will be available at https://bigtree765.github.io/C-Nav-project.

## 1 Introduction

Successful navigation to a target object [1, 2, 3, 4, 5] is a fundamental capability for embodied agents and serves as a prerequisite for any meaningful interaction, making it a central topic in recent research. Current state-of-the-art methods [6, 7, 8, 9, 10] typically depend on pre-trained models [11, 12, 13] and large-scale demonstration trajectories [7, 10], operating under the assumption of complete access to the training dataset and a fixed set of object categories. However, these approaches generally lack continual learning ability [14] and are vulnerable to catastrophic forgetting, which restricts their applicability in dynamic, open-world applications [15]. In contrast, practical deployment demands that agents adapt continuously to new object categories and evolving user instructions. Addressing this need requires agents to incrementally refine their navigation capabilities by learning from new data, without revisiting all previously seen data or retraining the model from scratch.

To enable the acquisition of new skills without forgetting previously learned knowledge, numerous studies have achieved promising results in uni-modal settings like image classification [16, 17].

---

†Project lead.    *Corresponding author.

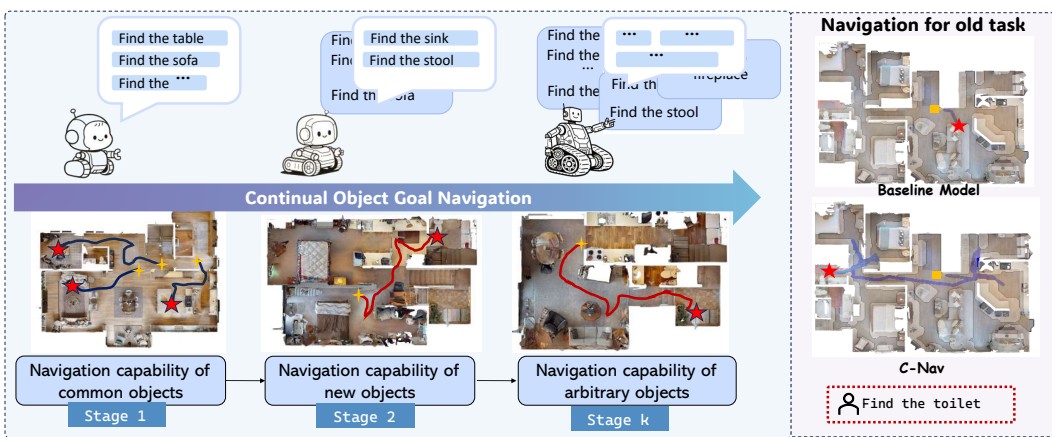

Figure 1: Continual Object Navigation: The robot must continually learn from new data while retaining its ability to navigate to previously seen object goals. After training across multiple phases, the baseline model suffers from *representation drift* and *policy degradation*, leading to a loss of navigation capability on previously learned tasks, whereas C-Nav enables cumulative knowledge retention across phases.

However, their application to embodied decision-making tasks, particularly object-goal navigation, remains largely unexplored. This might be because the task requires long-horizon decision-making and the integration of diverse multi-modal inputs, including RGB images, depth images, pose information, and textual instructions. Moreover, a single training trajectory can span hundreds of steps, which significantly increases the complexity of the learning process. To the best of our knowledge, continual learning in the context of object navigation has not been previously studied. To advance this important yet under-explored area, it is necessary to propose a comprehensive evaluation framework to assess the continual learning capabilities of object navigation models.

In related embodied navigation systems [6, 7, 8, 2, 18, 9, 10], the multi-modal encoder is often used for processing egocentric visual inputs, pose information, and other sensory signals to build a semantic representation of the environment. Simultaneously, the action decoder integrates temporal observations to predict navigation actions. As the agent is exposed to new tasks over time, distributional shifts in sensory inputs and action patterns introduce biases into the model components, leading to *representation drift* and *policy degradation*. This results in catastrophic forgetting of previously acquired knowledge, as illustrated in Figure 1. The agent often becomes capable of semantic exploration only for objects in the current task, losing its ability to navigate to goals learned earlier. Although naive data replay [19] has demonstrated potential in mitigating catastrophic forgetting, they face several notable limitations when applied to navigation tasks characterized by long sequences of temporally and spatially correlated observations: (1) storage overhead scales quickly with the number of tasks, making memory management increasingly costly; (2) retaining continuous scene observations raises privacy concerns, as these may inadvertently reveal sensitive spatial information; and (3) the reliance on fine-grained, atomic-level action representations results in substantial redundancy across adjacent frames, leading to inefficient memory usage and the replay of repetitive experiences.

To advance embodied AI research in realistic settings, we introduce a continual object navigation benchmark built upon the HM3D [20] and MP3D [21] datasets using the Habitat simulator [22]. This benchmark enables a comprehensive evaluation of mainstream navigation model architectures (including Bev-based [9], RNN-based [23], Transformer-based [18], and LLM-based models [8]) and several widely adopted continual learning techniques such as LoRA [24], model merge [25], LwF [26], and naive data replay [19]. Further, we propose C-Nav, a continual object navigation framework that incorporates a dual-path forgetting mitigation strategy. Specifically, a feature distillation path is employed to ensure representational consistency across multi-task, multi-modal encoders, while a feature replay path is designed to preserve decision stability within the policy decoder. This architectural design addresses the representational drift that occurs in both the encoder and policy modules as the agent encounters new tasks, while avoiding the storage overhead and privacy risks associated with raw trajectory retention. To mitigate the high visual redundancy inherent in navigation trajectories, C-Nav introduces an adaptive experience selection module. This module treats keyframe

selection as an outlier detection problem in the learned representation space and leverages the Local Outlier Factor (LOF) [27] algorithm to automatically identify critical navigation moments, such as "goal discovery" or "spatial transitions." In summary, our contributions include:

- To evaluate the continual learning ability of agents in dynamic task settings, we establish a continual object goal navigation benchmark, enabling a systematic assessment of state-of-the-art navigation architectures and continual learning strategies.

- We propose C-Nav, a continual Object Navigation framework that mitigates catastrophic forgetting via a dual-path strategy, enforcing representation consistency through feature distillation and policy consistency through feature replay, without relying on extensive raw trajectories.

- We develop an adaptive experience selection mechanism that identifies and retains informative features of navigation frames by leveraging representation-space outlier detection, significantly reducing memory redundancy while maintaining policy performance.

- Extensive experiments verify the effectiveness of C-Nav across architectures and continual learning methods, demonstrating superior performance in both accuracy and efficiency.

## 2 Related Work

### 2.1 Object Goal Navigation

The task of locating a specified object in an unknown environment is fundamental to robotic systems. Current approaches to object-goal navigation can be categorized into two types. (1) The first category focuses on Zero-Shot Object Navigation [28, 29, 4, 3, 30]. These methods explicitly construct maps and utilize VLMs [31, 32, 33, 34] or LLMs [35, 36, 37] for reasoning to select the most valuable frontier or waypoint for exploration. For instance, in Cow [29], the robot explores the nearest frontier point until the target is detected using CLIP features [11] and open-vocabulary object detectors [13]. ESC [5], L3MVN [38], and Voronav [3] leverage LLMs for reasoning and decision-making to enhance performance. VLFM [4] employs a VLM to assign semantic values to the map based on first-person observations and textual prompts, selecting the highest-scoring frontier. These approaches exhibit strong zero-shot generalization and avoid catastrophic forgetting, as they do not update model parameters. However, they rely on complex reasoning pipelines and manually designed exploration rules, making them cumbersome and computationally costly. Moreover, their inability to fine-tune on newly collected navigation data limits task-specific adaptation, often resulting in suboptimal performance in complex or novel environments. (2) The second category involves using pre-trained visual encoders to transform first-person observations into visual vectors, which are then processed by a navigation policy trained via large-scale imitation or reinforcement learning [6, 7, 8, 2, 18, 9, 10, 39, 40], thereby equipping agents with semantic navigation capabilities.

These approaches, often trained via imitation or supervised learning, provide robust performance for object-goal navigation tasks by incorporating large-scale datasets. However, these methods typically assume fixed environments and task categories, lacking the ability to learn continually. As a result, they struggle to adapt to dynamic settings where new objects and tasks emerge over time and are prone to catastrophic forgetting, which limits their generalization to open-world scenarios. Recent advances such as FSTTA [41] and NavMorph [42] enhance test-time adaptability through fast–slow gradient updates or self-evolving world models, but primarily focus on fine-grained, step-by-step navigation. In contrast, our work addresses coarse-grained, long-horizon navigation, where the agent interprets high-level object-goal instructions to perform long-range exploration. Moreover, unlike prior methods that train on static datasets, our framework progressively introduces new object categories across multiple training stages, enabling continual skill acquisition. To this end, we propose C-Nav, a continual navigation framework that equips agents with the ability to incrementally learn new capabilities and adapt to novel settings without forgetting previously acquired knowledge.

### 2.2 Continual Learning

Continual learning [14, 43, 44], also known as incremental learning or lifelong learning, has attracted significant attention in recent years. Existing methods mainly focus on image classification. Regularization methods [26, 43] aim to prevent catastrophic forgetting by constraining the changes in model parameters during learning. Data replay methods [45, 46], which store and reuse old data, can retain

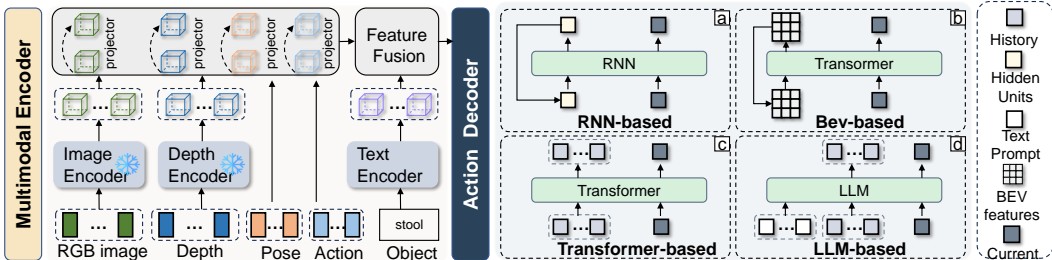

Figure 2: Different navigation model architectures. According to the action decoder, they can be categorized into a) RNN-Based, b) Bev-Based, c) Transformer-Based, and d) LLM-Based.

prior knowledge but introduce additional storage and privacy concerns [47]. Architecture-based methods [48] add new submodules for each task, leading to an increase in model parameters as the learning progresses. Recently, some studies [49, 50, 51, 52] have explored continual learning of multimodal large language models with parameter-efficient fine-tuning techniques (e.g., LoRA [24, 53], prompt tuning [54]) to facilitate continual fine-tuning without significant computational overhead. However, these methods do not extend well to embodied intelligence, especially in long-range, multimodal ObjectNav tasks. In contrast, embodied navigation poses greater challenges, requiring agents to process multimodal inputs and adapt to evolving environments and goals. Existing continual learning techniques fall short in addressing such complexity, particularly in long-horizon, multimodal tasks demanding both spatial and semantic reasoning. This gap motivates our C-Nav framework, which enables continual learning for visual navigation, with a focus on long-horizon tasks, multimodal integration, and dynamic adaptation in real-world settings.

## 3 Problem Definition and Benchmark

**Problem Setting.** We propose *Continual-ObjectNav*, where an embodied agent must incrementally master navigation skills through $K$ sequential tasks. Each task $k \in \{1, \dots, K\}$ introduces a distinct set of goal object categories $\mathcal{C}_k$, with strict disjointness between tasks ($\mathcal{C}_i \cap \mathcal{C}_j = \emptyset$ for all $i \neq j$). The agent receives training data $\mathcal{D}_k = \{(s_i^k, c_i^k, \tau_i^k)\}_{i=1}^{N_k}$ at each stage, where $s_i^k$ denotes a 3D indoor environment, $c_i^k \in \mathcal{C}_k$ specifies the target object category, and $\tau_i^k = \{(o_{i,t}^k, a_{i,t}^k)\}_{t=1}^{L_i}$ provides expert demonstration trajectories. The observation $o_{i,t}^k$ integrates egocentric RGB-D sensing, agent pose (position/orientation), action history, and object name. The action $a_{i,t}^k$ is selected from a discrete action space $\mathcal{A} = \{\texttt{move\_forward}, \texttt{turn\_left}, \texttt{turn\_right}, \texttt{look\_up}, \texttt{look\_down}, \texttt{stop}\}$.

**Evaluation.** We evaluate the agent at the end of each training stage $k$, using a test set of unseen environments $\mathcal{S}_k$ and object goals sampled from the cumulative category set $\mathcal{C}_{1:k} = \bigcup_{j=1}^{k} \mathcal{C}_j$. This setup requires the agent to retain and apply previously learned navigation skills while adapting to new categories. Each episode starts from a random position, and the agent must navigate to a target object category $c \in \mathcal{C}_{1:k}$. A success is recorded if the agent issues a $\texttt{stop}$ action within a specified distance to a valid instance of the target object within 500 time steps. We report the most widely used metrics: Success Rate (SR) and Success weighted by Path Length (SPL), averaged across all categories in $\mathcal{C}_{1:k}$. SPL measures the efficiency of the agent's trajectory by comparing it to the shortest possible path from the starting position to the nearest valid instance of the target object category.

**Model Architecture Notation.** As illustrated in Figure 2, the agent policy, shared across different architectures, can be parameterized as a composition of two learnable modules: a multimodal encoder $f : \mathcal{O} \rightarrow \mathbb{R}^d$, which maps observations to compact feature representations $h_t = f(o_t)$, and an action decoder $\pi$ that predicts navigation actions conditioned on both the current encoded feature and the trajectory history. At each time step $t$, the decoder receives $h_{1:t} = \{f(o_1), \dots, f(o_t)\}$ and produces a distribution over actions $\pi(a_t|h_{1:t})$.

**Dataset.** We adopt two widely used object goal navigation datasets: ObjectNav (HM3D) consists of 2,000 episodes sampled from 20 validation scenes in the HM3D dataset, covering 6 object categories. ObjectNav (MP3D), introduced in the Habitat 2020 Challenge, contains 2,195 episodes from 11 MP3D validation scenes, spanning 21 object categories. For model training, we utilize human demonstration trajectories collected via Habitat-Web [10] and PIRL [7]. The HM3D training set contains 75,488 trajectories, while the MP3D training set includes 59,604 trajectories after filtering

Table 1: Continual-ObjectNav splits for MP3D and HM3D.

| Type | MP3D | | | | | HM3D | | | | |
|---|---|---|---|---|---|---|---|---|---|---|
| | Stage1 | Stage2 | Stage3 | Stage4 | Total | Stage1 | Stage2 | Stage3 | Stage4 | Total |
| Category | 12 | 3 | 3 | 3 | 21 | 3 | 1 | 1 | 1 | 6 |
| Trajectory (training data) | 40,608 | 7,306 | 7,484 | 4,206 | 59,604 | 36,667 | 13,640 | 13,182 | 11,999 | 75,488 |
| Episode (evaluation data) | 1,783 | 1,934 | 2,039 | 2,195 | 2,195 | 945 | 1,343 | 1,624 | 2,000 | 2,000 |

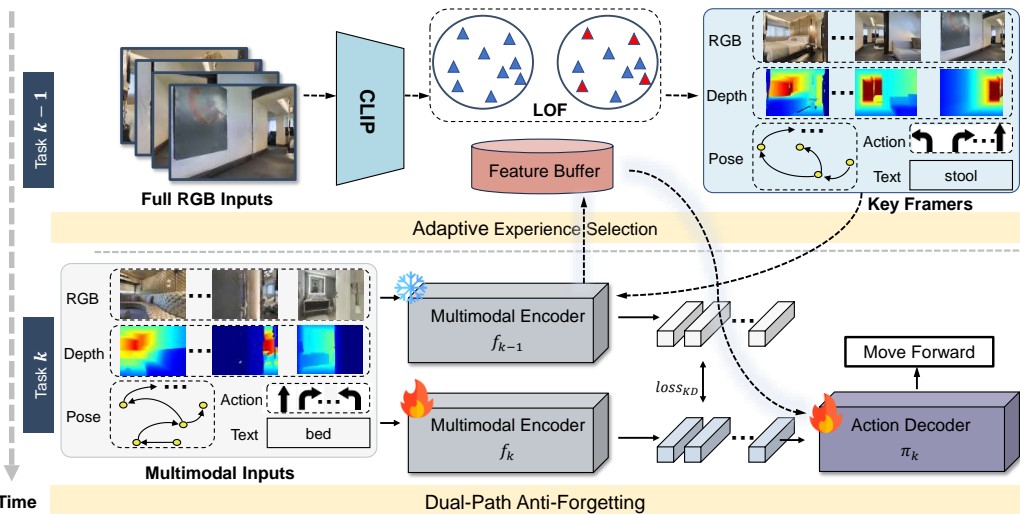

Figure 3: Overview of the proposed **C-Nav** framework for continual object navigation. It consists of two key components: (1) adaptive experience selection, which identifies semantically meaningful keyframes via LOF in the representation space, reducing storage and redundancy; and (2) dual-path anti-forgetting, which mitigates representation and policy drift through a feature distillation path (bottom left) and a feature replay path (bottom right), ensuring stability across encoder and decoder modules. For each task, C-Nav retains $p$ sparse trajectory features in the deep feature space.

out samples involving 6 synthetic objects. To adapt to the continual Object Navigation setting, we divide the object categories and corresponding trajectories into four incremental learning stages. The detailed splits are shown in Table 1, and a full list of object category assignments per stage can be found in the supplementary materials.

## 4   The Proposed Method: C-Nav

To address the continual object navigation task more effectively, we propose a new method described in this section. The overall framework of our approach is illustrated in Figure. 3.

### 4.1   Dual-Path Anti-Forgetting

In continual learning of object navigation, catastrophic forgetting can be attributed to representational drift in both the multimodal encoder and the action decoder. As the model sequentially learns new tasks, the distribution of learned features may shift, resulting in degraded performance on earlier tasks. To address this, we propose a dual-path anti-forgetting strategy that constrains both the feature extraction and policy prediction stages.

**Representation Consistency with Feature Distillation.** The multimodal encoder integrates visual and goal-related information. To prevent drift in the learned feature representation, we apply feature-level distillation. Specifically, we enforce the consistency between the old and current encoder outputs by minimizing the $\ell_2$ distance between their extracted features across time:

$$\mathcal{L}_{\mathrm{KD}} = \sum_{t=1}^{L} \|f_{k-1}(o_t) - f_k(o_t)\|_2^2, \tag{1}$$

where $f_{k-1}$ denotes the frozen encoder from a previous task, and $f_k$ is the current encoder. Both take as input the observation $o_t$, ensuring multimodal alignment is preserved during continual learning.

**Policy Consistency with Feature Replay.** To mitigate the policy forgetting during continual learning, we replay stored trajectory features of previous tasks along with their corresponding action labels to maintain policy consistency. The feature replay loss is defined as:

$$\mathcal{L}_{\text{FR}} = \frac{1}{|L|} \sum_{t=1}^{L} -w_t \log \pi_k(a_t|h_{1:t}), \quad w_t = 1 + \gamma \cdot \mathbb{1}_{a_t \neq a_{t-1}}, \tag{2}$$

where $h_t \in \mathbb{R}^d$ represents the $t$-th frame feature stored in the feature buffer, and $\pi_k$ denotes the current action decoder. The weighting factor $\gamma$ [10] emphasizes action transitions by assigning higher importance to time steps where the expert action differs from the previous one. This objective function ensures that the policy maintains consistent interpretations of previously learned features while adapting to new tasks. The gradient updates through this loss term preserve the mapping between historical observations and their originally learned actions, thereby mitigating catastrophic forgetting in the continual learning scenario.

## 4.2 Adaptive Experience Selection

In continual object navigation, replaying previously collected trajectories is critical for mitigating catastrophic forgetting. However, storing all frames in a trajectory is memory-intensive and often wasteful, as many frames are visually redundant and contain little novel information. From a data distribution perspective, frames that exhibit significant semantic shifts tend to be rare and thus can be considered outliers. These frames often correspond to moments such as entering new spaces, approaching target objects, or reaching decision points. Therefore, we propose to transform the task of selecting salient experience points into the detection of outliers within the feature space of visual observations. Specifically, given a trajectory $\tau = \{o_1, o_2, \ldots, o_L\}$, we first encode visual observations using a pretrained CLIP model to obtain feature representations: $\mathbf{v}_i = \text{CLIP}(\text{RGB}(o_i))$ for $i = \{1, \ldots, L\}$. Let $\text{dist}(\mathbf{v}_i, \mathbf{v}_j)$ denote the Cosine distance in feature space:

$$d(\mathbf{v}_i, \mathbf{v}_j) = 1 - \frac{\mathbf{v}_i^\top \mathbf{v}_j}{\|\mathbf{v}_i\|_2 \cdot \|\mathbf{v}_j\|_2}. \tag{3}$$

Base that, we define the $k$-distance neighborhood: $N_k(\mathbf{v}_i) = \{\mathbf{v}_j \mid d(\mathbf{v}_i, \mathbf{v}_j) \leq d_k(\mathbf{v}_i)\}$, where $d_k(\mathbf{v}_i)$ is the distance to the $k$-th nearest neighbor. For each neighbor $\mathbf{v}_j \in N_k(\mathbf{v}_i)$, the reachability distance is defined as $\text{reach-dist}_k(\mathbf{v}_i, \mathbf{v}_j) = \max\{d_k(\mathbf{v}_j), d(\mathbf{v}_i, \mathbf{v}_j)\}$. Using this, the local reachability density (LRD) of $\mathbf{v}_i$ is the inverse of its mean reachability distance:

$$\text{LRD}_k(\mathbf{v}_i) = \left( \frac{1}{|N_k(\mathbf{v}_i)|} \sum_{\mathbf{v}_j \in N_k(\mathbf{v}_i)} \text{reach-dist}_k(\mathbf{v}_i, \mathbf{v}_j) \right)^{-1}. \tag{4}$$

Then, the LOF is calculated as the relative density of $\mathbf{v}_i$ compared to its neighbors:

$$\text{LOF}_k(\mathbf{v}_i) = \frac{1}{|N_k(\mathbf{v}_i)|} \sum_{\mathbf{v}_j \in N_k(\mathbf{v}_i)} \frac{\text{LRD}_k(\mathbf{v}_j)}{\text{LRD}_k(\mathbf{v}_i)}. \tag{5}$$

According to this formulation, the set of selected keyframe indices is defined as $\mathcal{I} = \{i \mid \text{LOF}_k(\mathbf{v}_i) > 1\}$, where a higher $\text{LOF}_k(\mathbf{v}_i)$ indicates that the frame is considered as an outlier, typically corresponding to a navigation state with significant semantic change. These keyframes are subsequently encoded through a multimodal encoder $f_k$ and stored in a feature buffer to mitigate catastrophic forgetting in the action decoder during training.

## 4.3 Overall Learning Objective of C-Nav

Our continual navigation policy is optimized using a composite loss function that integrates supervision from the current task, feature-space knowledge distillation, and feature replay consistency:

$$\mathcal{L} = \mathcal{L}_{\text{Curr}} + \lambda_{\text{KD}} \cdot \mathcal{L}_{\text{KD}} + \lambda_{\text{FR}} \cdot \mathcal{L}_{\text{FR}}, \tag{6}$$

where $\lambda_{\text{KD}}$ and $\lambda_{\text{FR}}$ are weighting coefficients that balance the influence of feature-level knowledge distillation and feature replay loss, respectively. The current task $k$ using behavior cloning with inflection weighting [10]: $\mathcal{L}_{\text{curr}} = \frac{1}{L} \sum_{t=1}^{L} -w_t \log \pi_k \left( a_t | f_k(o_1), \ldots, f_k(o_t) \right)$.

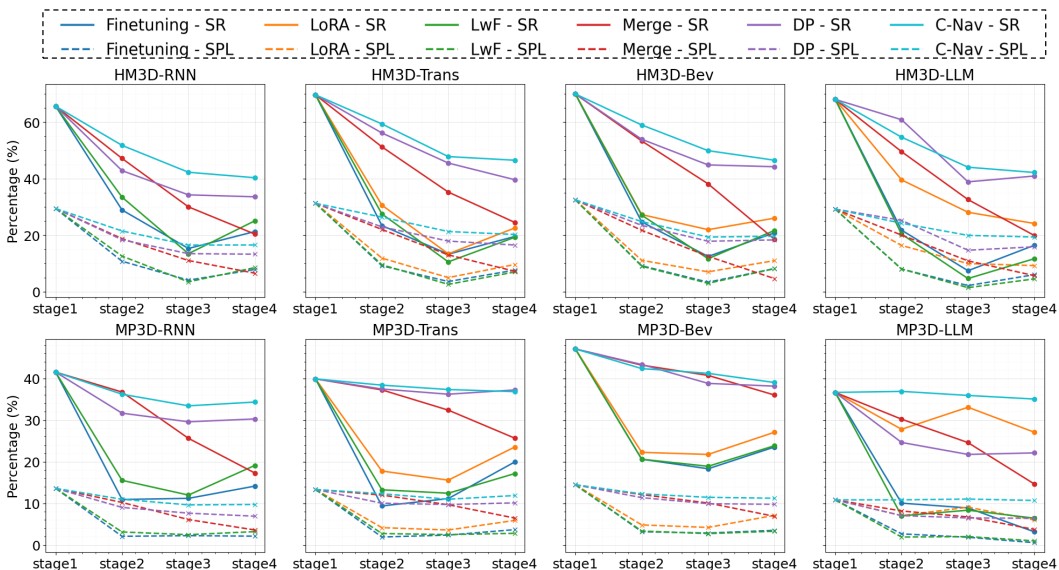

Figure 4: Results of SR and SPL on the MP3D and HM3D datasets. Solid lines represent SR, while dashed lines represent SPL.

Table 2: Performance comparison on the HM3D benchmark. *DP* indicates data replay, and *Merge* denotes model merging. *Avg* reports the average performance across all stages, while *Last* reflects the performance at the final stage.

| Methods | HM3D-RNN | | | | HM3D-Trans | | | | HM3D-Bev | | | | HM3D-LLM | | | |
|---|---|---|---|---|---|---|---|---|---|---|---|---|---|---|---|---|
| | Avg | | Last | | Avg | | Last | | Avg | | Last | | Avg | | Last | |
| | SR | SPL | SR | SPL | SR | SPL | SR | SPL | SR | SPL | SR | SPL | SR | SPL | SR | SPL |
| Finetuning | 32.8 | 13.0 | 21.3 | 7.8 | 31.4 | 12.9 | 19.5 | 7.6 | 32.0 | 13.3 | 20.8 | 8.2 | 28.4 | 11.3 | 16.4 | 6.0 |
| LoRA | - | - | - | - | 34.0 | 14.4 | 22.5 | 9.6 | 36.3 | 15.4 | 24.1 | 9.2 | 39.9 | 16.2 | 24.1 | 9.2 |
| LwF | 34.4 | 13.5 | 25.1 | 8.5 | 31.7 | 12.6 | 19.2 | 6.9 | 32.6 | 13.1 | 21.7 | 8.1 | 26.2 | 10.8 | 11.7 | 4.5 |
| Merge | 40.8 | 16.4 | 20.4 | 6.5 | 45.1 | 18.4 | 24.5 | 7.2 | 45.0 | 17.8 | 18.5 | 4.6 | 42.5 | 16.4 | 19.9 | 5.6 |
| DP | 44.1 | 18.6 | 33.6 | 13.2 | 52.7 | 22.1 | 39.6 | 16.5 | 53.2 | 23.0 | 44.2 | 18.3 | 52.2 | 21.2 | 40.9 | 15.9 |
| C-Nav | **50.0** | **21.0** | **40.3** | **16.5** | **55.8** | **24.8** | **46.5** | **20.2** | **56.3** | **24.0** | **46.5** | **19.6** | **52.2** | **23.1** | **42.2** | **19.3** |

# 5  Experiments

## 5.1  Experimental Setup

**Implementation Details.** We implement our multi-modal encoder using CLIP-ResNet50 [11] for visual encoding and a PointNav-pretrained ResNet-50 [55] for depth encoding. Following previous work [23, 9], we keep both encoders frozen during training. The feature fusion is implemented by the feature concatenation. We train our model using AdamW optimization with a linear warmup over 1,000 steps to reach our initial learning rate of $3 \times 10^{-4}$, followed by linear decay. We train each task stage for 25 epochs with a batch size of 32. For action prediction, the RNN-based model employs a 2-layer LSTM [56] architecture. The transformer-based and Bev-based [9] decoders utilize a 4-layer transformer [57] as the action decoder with a dropout rate of 0.1. For the LLM-based approach, we adopt Qwen2-0.5B [37] as the action decoder, incorporating six special tokens to represent atomic actions. We set the inflection weight $\gamma$ to 3.48 and configure our loss balance weights $\lambda_{\text{KD}}$ and $\lambda_{\text{FP}}$ to 5. All experiments are conducted using two NVIDIA A6000 GPUs. Additional implementation details are provided in the supplementary materials.

**Comparison Methods.** We compare our method against several representative continual learning strategies across four backbone architectures. Specifically, we evaluate the following baselines. Naive Fine-tuning serves as the simplest approach, where models are sequentially fine-tuned on each stage without any mechanism to mitigate forgetting. LoRA [24] introduces low-rank adaptation modules during training; after each stage, the LoRA branches are merged into the main model for inference. LwF [26] applies knowledge distillation by computing the KL divergence between the logits of the

Table 3: Performance comparison of different methods on MP3D benchmark.

| Methods | MP3D-RNN | | | | MP3D-Trans | | | | MP3D-Bev | | | | MP3D-LLM | | | |
|---|---|---|---|---|---|---|---|---|---|---|---|---|---|---|---|---|
| | Avg | | Last | | Avg | | Last | | Avg | | Last | | Avg | | Last | |
| | SR | SPL | SR | SPL | SR | SPL | SR | SPL | SR | SPL | SR | SPL | SR | SPL | SR | SPL |
| Finetuning | 19.4 | 5 | 14.1 | 2.1 | 20.1 | 5.3 | 19.9 | 3.7 | 27.4 | 6.0 | 23.5 | 3.5 | 14.7 | 4.0 | 3.1 | 0.5 |
| LoRA | - | - | - | - | 24.2 | 6.7 | 23.5 | 5.9 | 29.5 | 7.6 | 27.1 | 7.1 | 31.1 | 8.2 | 27.1 | 6.0 |
| LwF | 22 | 5.6 | 19.1 | 3.2 | 20.7 | 5.3 | 17.1 | 2.8 | 27.6 | 5.9 | 23.8 | 3.3 | 14.6 | 3.9 | 6.4 | 1.0 |
| Merge | 30.3 | 8.4 | 17.3 | 3.6 | 33.8 | 10.3 | 25.7 | 6.5 | 41.7 | 10.9 | 36.0 | 6.9 | 26.5 | 7.4 | 14.6 | 3.7 |
| DP | 33.8 | 9.3 | 30.3 | 6.9 | 37.7 | 10.8 | 37.2 | 10.1 | 41.8 | 11.4 | 38.1 | 9.8 | 26.3 | 7.7 | 22.1 | 6.4 |
| C-Nav | **36.4** | **11.0** | **34.3** | **9.7** | **38.1** | **12.1** | **36.8** | **11.9** | **42.4** | **12.3** | **39.0** | **11.2** | **36.1** | **10.8** | **35.0** | **10.7** |

current model and those of the previous one, encouraging the model to preserve past knowledge. Model merge preserves prior knowledge by linearly interpolating the weights of the current and previous models using a 0.7/0.3 ratio. Lastly, data replay stores $p = 80$ original trajectories per object category, which are reused in subsequent training stages to mitigate forgetting. Additional implementation details are provided in the supplementary materials.

## 5.2 Experimental Results and Analysis

**Main Results.** We compare several baseline methods and model architectures for Continual-ObjectNav task. As shown in Figure 4, naive fine-tuning leads to severe catastrophic forgetting across all four architectures, with an average decline of approximately 40% in navigation success rate on previously seen categories by the final stage. Additionally, the results show that the Bev-based architecture consistently outperforms others. Meanwhile, LLM-based models do not exhibit a clear performance advantage, likely due to limited training trajectories. The LwF method demonstrates limited effectiveness across all models. In contrast, LoRA and model merge help mitigate forgetting by either reducing the number of trainable parameters or integrating knowledge from previous models. Notably, LoRA shows a significant advantage when applied to LLM-based models compared to other model architectures. Data replay achieves effective performance retention, but it incurs substantial storage overhead and raises privacy concerns. As shown in Table 2 and Table 3, the proposed C-Nav generalizes well across various architectures and datasets. Specifically, compared to data replay, C-Nav achieves an average SR improvement of 3.35% on MP3D and 2.75% on HM3D across four model architectures, while requiring lower storage costs. A detailed breakdown of performance on old versus new tasks at each training stage is provided in Appendix C.1, illustrating how C-Nav balances retention of prior knowledge with adaptation to new tasks. These results demonstrate the effectiveness of our dual-path forgetting mitigation mechanism, which ensures consistent representation and policy learning as the agent encounters new tasks.

Table 4: Ablation studies of dual-path anti-forgetting. KD denotes feature distillation, and FP denotes feature replay.

| Methods | HM3D-RNN | | | | HM3D-Trans | | | | HM3D-Bev | | | | HM3D-LLM | | | |
|---|---|---|---|---|---|---|---|---|---|---|---|---|---|---|---|---|
| | Avg | | Last | | Avg | | Last | | Avg | | Last | | Avg | | Last | |
| | SR | SPL | SR | SPL | SR | SPL | SR | SPL | SR | SPL | SR | SPL | SR | SPL | SR | SPL |
| w/o KD | 28.2 | 11.7 | 16.9 | 6.6 | 31.4 | 13.2 | 20.2 | 8.1 | 32.5 | 13.9 | 21.9 | 8.9 | 33.6 | 13.4 | 19.9 | 7.3 |
| w/o FP | 37.9 | 15.6 | 27.9 | 10.7 | 45.9 | 21.9 | 32.6 | 14.5 | 38.9 | 16.5 | 26.7 | 10.3 | 42.7 | 17.2 | 30.2 | 11.6 |
| All | **50.0** | **21.0** | **40.3** | **16.5** | **55.8** | **24.8** | **46.5** | **20.2** | **56.3** | **24.0** | **46.5** | **19.6** | **52.2** | **23.1** | **42.2** | **19.3** |

| Methods | MP3D-RNN | | | | MP3D-Trans | | | | MP3D-Bev | | | | MP3D-LLM | | | |
|---|---|---|---|---|---|---|---|---|---|---|---|---|---|---|---|---|
| | Avg | | Last | | Avg | | Last | | Avg | | Last | | Avg | | Last | |
| | SR | SPL | SR | SPL | SR | SPL | SR | SPL | SR | SPL | SR | SPL | SR | SPL | SR | SPL |
| w/o KD | 16.9 | 4.7 | 10.4 | 2.0 | 20.6 | 5.8 | 21.2 | 4.6 | 26.1 | 6.6 | 23.5 | 5.2 | 25.7 | 6.4 | 24.0 | 4.8 |
| w/o FP | 25.9 | 8.1 | 22.0 | 7.3 | 27.2 | 8.9 | 22.7 | 8.5 | 31.3 | 9.6 | 29.2 | 9.95 | 28.9 | 9.0 | 24.7 | 8.0 |
| All | **36.4** | **11.0** | **34.3** | **9.7** | **38.1** | **12.1** | **36.8** | **11.9** | **42.4** | **12.3** | **39.0** | **11.2** | **36.1** | **10.8** | **35.0** | **10.7** |

**Ablation Studies of Dual-Path Anti-forgetting.** To assess the contribution of each component in our proposed dual-path anti-forgetting mechanism, we conduct ablation studies by separately removing

the feature-space alignment loss and the feature replay loss in the action decoder. As presented in Table 4 and Appendix C.4, the removal of either component leads to significant performance degradation. Specifically, eliminating the feature-space KD loss results in an average SR drop of 22% and 16% across four model architectures on the HM3D and MP3D datasets, respectively. Similarly, removing the feature replay loss from the action decoder causes a reduction of 12% and 10% on the same datasets. These results highlight the complementary roles of the two components: the feature-space constraint stabilizes representation learning, while feature replay ensures consistent decision-making under distributional shifts. Additionally, we further examine representation consistency in continual learning. As detailed in Appendix C.2, we compare fully freezing the multimodal encoder against selectively fine-tuning only the modality-specific projectors with consistency regularization to softly align old and new feature spaces. The results show that fully freezing the encoder degrades performance, whereas selective fine-tuning of projectors under consistency constraints enables the model to learn representations that remain effective for both previous and new tasks.

**Ablation Studies of Adaptive Experience Selection.** To validate the effectiveness of the proposed adaptive experience selection in C-Nav, we compare it with a uniform sampling baseline that assumes redundancy among adjacent frames. For a fair comparison, both methods replay trajectories truncated to 50% of their original length. As shown in Table 5 and Appendix C.5, our method consistently outperforms uniform sampling across model architectures and datasets, achieving SR improvements of 3.65% on HM3D and 3.2% on MP3D. Notably, compared to full-length feature replay in C-Nav, SR drops only marginally by 1.9% and 1.3%, despite using half the data. In addition, compared to full data replay, our adaptive method achieves an average SR improvement of 0.8% on HM3D and 2.0% on MP3D across four model architectures. These results demonstrate the efficiency and effectiveness of our adaptive strategy in preserving critical experiences for the Continual-ObjectNav task.

Table 5: Ablation studies of adaptive experience selection. "Uniform", "Adaptive", and "Full" refer to uniform sampling, adaptive sampling, and no sampling, respectively.

| Methods | HM3D-RNN | | | | HM3D-Trans | | | | HM3D-Bev | | | | HM3D-LLM | | | |
| | Avg | | Last | | Avg | | Last | | Avg | | Last | | Avg | | Last | |
| | SR | SPL | SR | SPL | SR | SPL | SR | SPL | SR | SPL | SR | SPL | SR | SPL | SR | SPL |
| Uniform | 43.2 | 19.7 | 32.2 | 15.0 | 50.5 | 22.0 | 40.5 | 17.6 | 49.4 | 21.7 | 37.6 | 16.5 | 47.7 | 22.4 | 37.0 | 18.4 |
| DP (Full) | 44.1 | 18.6 | 33.6 | 13.2 | 52.7 | 22.1 | 39.6 | 16.5 | 53.2 | 23.0 | 44.2 | 18.3 | **52.2** | 21.2 | 40.9 | 15.9 |
| Adaptive | 47.3 | 20.7 | 36.1 | 16.0 | 53.7 | 23.2 | 42.5 | 18.4 | 52.8 | 22.4 | 42.1 | 18.0 | 51.6 | **23.8** | 40.1 | **19.4** |
| Full | **50.0** | **21.0** | **40.3** | **16.5** | **54.7** | **24.3** | **46.5** | 20.2 | **56.3** | **24.0** | **46.5** | **19.6** | **52.2** | 23.1 | **42.2** | 19.3 |

| Methods | MP3D-RNN | | | | MP3D-Trans | | | | MP3D-Bev | | | | MP3D-LLM | | | |
| | Avg | | Last | | Avg | | Last | | Avg | | Last | | Avg | | Last | |
| | SR | SPL | SR | SPL | SR | SPL | SR | SPL | SR | SPL | SR | SPL | SR | SPL | SR | SPL |
| Uniform | 30.5 | 10.3 | 24.5 | 8.2 | 35.2 | 11.4 | 31.9 | 9.7 | 35.0 | 11.0 | 29.2 | 9.7 | 34.0 | 11.1 | 31.4 | **10.9** |
| DP (Full) | 33.8 | 9.3 | 30.3 | 6.9 | 37.7 | 10.8 | 37.2 | 10.1 | 41.8 | 11.4 | 38.1 | 9.8 | 26.3 | 7.7 | 22.1 | 6.4 |
| Adaptive | 34.1 | **11.4** | **38.4** | **10.1** | **38.4** | **12.6** | 36.6 | 11.7 | 40.9 | **12.3** | 38.8 | **11.3** | 34.2 | **11.2** | 32.4 | 10.7 |
| Full | **36.4** | 11.0 | 34.3 | 9.7 | 38.1 | 12.1 | **36.8** | **11.9** | **42.4** | **12.3** | **39.0** | 11.2 | **36.1** | 10.8 | **35.0** | 10.7 |

Moreover, Figure 5 presents an ablation study of different experience selection strategies on MP3D-Trans across all training stages. First, we compare our method against uniform sampling and clustering-based sampling, both using half-length trajectories. Our approach consistently outperforms both baselines because it explicitly pre-

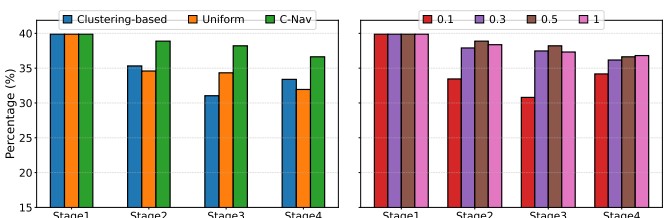

Figure 5: Ablation on experience selection and retention ratio on MP3D with a Transformer-based navigation architecture.

serves semantically salient frames, such as those captured at decision points or near target objects. These frames often reside in outlier regions of the feature space and are typically discarded by conventional samplers. Second, we reduce the key-frame selection ratio. Remarkably, our method

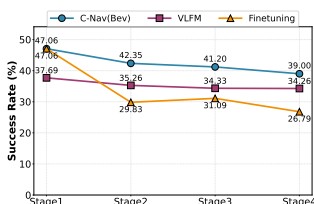

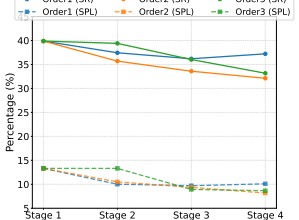

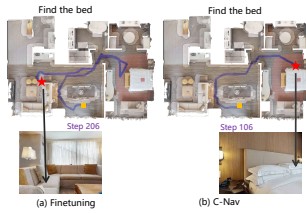

Figure 6: Comparison with the Zero-Shot Method.

Figure 7: Sensitivity to stage order on MP3D.

Figure 8: Qualitative navigation results.

maintains strong performance even at very low retention rates, achieving comparable or better results while using only 20% of the memory required by the clustering-based or uniform sampling baselines. This demonstrates the high efficiency and robustness of our approach.

**Comparison with the Zero-Shot Method.** We reproduce VLFM as a zero-shot baseline and evaluate it against C-Nav over four continual learning stages in Figure 6. VLFM selects waypoints using a pretrained vision-language model on an explicit map and executes navigation with a fixed reinforcement learning policy. Because its parameters are frozen, it avoids catastrophic forgetting and maintains stable but limited performance. This limitation stems from its inability to learn from new trajectories, leaving it constrained by the static priors of the large model. In contrast, C-Nav is an end-to-end trainable system that continuously improves from experience, adapts to new tasks, and mitigates forgetting.

**Ablation Study on Stage Order Sensitivity.** To evaluate the robustness of C-Nav to variations in task sequence during continual learning, we conduct ablation studies under three different stage orderings. As shown in Figure 7, the minimal performance variation across orderings further validates the effectiveness of our Dual-Path Anti-Forgetting mechanism. This insensitivity to task sequence is crucial for real-world deployment, where object categories may be encountered in arbitrary order.

**Case Study.** Figure 8 shows navigation trajectories of C-Nav and the Finetuning baseline in an unseen HM3D scene, where the target object (bed) was introduced in an earlier training stage. Both models were trained sequentially over four stages, with the sofa added as the target in the final stage. The baseline suffers from catastrophic forgetting and navigates toward a sofa despite the goal being a bed. In contrast, C-Nav successfully reaches the bed, demonstrating its ability to retain knowledge of previously learned categories. This highlights C-Nav's superior stability-plasticity balance in continual learning.

# 6 Conclusion

In this work, we establish a benchmark for evaluating continual learning in the context of object navigation. We assess various model architectures and mainstream continual learning methods. Building on this benchmark, we propose C-Nav for continual object navigation, a dual-path framework that mitigates catastrophic forgetting by jointly enforcing representation consistency through feature distillation and preserving policy consistency via feature replay. Additionally, we introduce an adaptive experience selection method to further reduce memory usage, ensuring efficient knowledge retention. Extensive experiments demonstrate that our approach outperforms existing methods, achieving superior performance across different architectures and tasks, while reducing memory overhead and mitigating forgetting more effectively than previous solutions.

**Limitations.** Despite the strong performance of C-Nav on the Continual-ObjectNav task, several limitations remain. First, real-world factors such as dynamic lighting and sensor noise may affect generalization, highlighting the need for validation on physical robots to assess robustness in real environments. Second, although C-Nav significantly reduces memory usage, its storage demands still grow linearly with task complexity. Future work may explore generative replay or trajectory-free approaches to further reduce memory overhead. Addressing these limitations is crucial for deploying continual navigation systems in practical applications.

## Acknowledgments and Disclosure of Funding

This research is supported by the Artificial Intelligence National Science and Technology Major Project (2023ZD0121200), the National Natural Science Foundation of China (62437001, 62436001, 62206279), the Key Research and Development Program of Jiangsu Province (BE2023016-3), the Beijing Natural Science Foundation (L252146), and the InnoHK program.

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

# A Dataset Details

## A.1 Category Splits for Continual ObjectNav

**HM3D Dataset.** The object categories are split into four stages as follows:

- **Stage 1:** *bed*, *chair*, *plant*
- **Stage 2:** *toilet*
- **Stage 3:** *tv_monitor*
- **Stage 4:** *sofa*

**MP3D Dataset.** The object categories are divided into four continual learning stages:

- **Stage 1:** *bed*, *cabinet*, *chair*, *chest_of_drawers*, *cushion*, *picture*, *plant*, *sink*, *sofa*, *stool*, *table*, *toilet*
- **Stage 2:** *shower*, *towel*, *tv_monitor*
- **Stage 3:** *bathtub*, *counter*, *fireplace*
- **Stage 4:** *clothes*, *gym_equipment*, *seating*

## A.2 Demonstration Trajectory Length Distribution for Continual ObjectNav

We visualize the trajectory length distributions across different training stages in Continual ObjectNav, as shown in Figure 9 and Figure 10. For the HM3D dataset, the average trajectory lengths for Stage 1 to Stage 4 are 127.51, 142.53, 168.53, and 115.77, respectively. In contrast, the MP3D dataset exhibits longer trajectories, with average lengths of 182.79, 247.91, 227.10, and 315.75 across the four stages. The longer demonstration sequences in MP3D highlight the increased complexity of decision-making, making the Continual ObjectNav task more challenging.

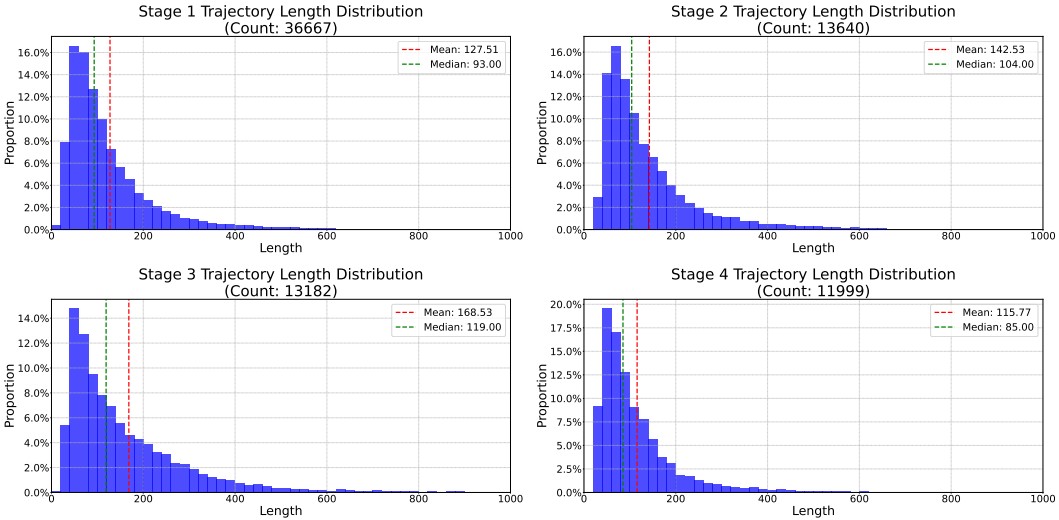

Figure 9: Demonstration trajectory length distribution per stage in HM3D dataset.

# B Implementation Details

## B.1 Details about the Model Architecture

For the BEV-based, transformer-based, and RNN-based models, the RGB, depth, pose, previous action, and object name inputs are transformed by linear projection layers into feature vectors with dimensionalities of 256, 128, 64, 32, and 32, respectively. In the Qwen 0.5B LLM-based architecture,

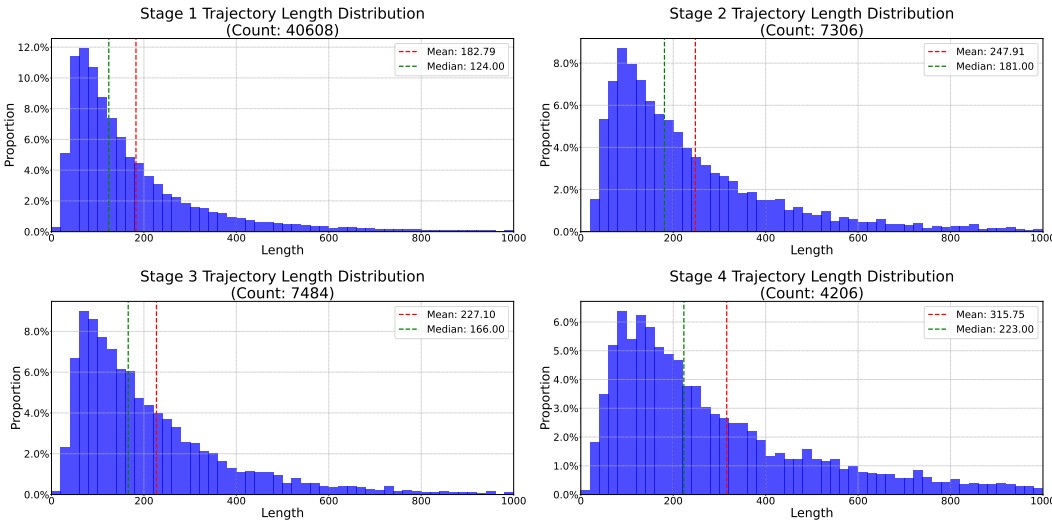

Figure 10: Demonstration trajectory length distribution per stage in MP3D Dataset.

the RGB, depth, pose, and previous action inputs are projected into feature vectors of size 400, 400, 64, and 32, respectively. These vectors are concatenated to produce a unified representation with a total dimensionality of 896. The task prompt used for the LLM-based model is formatted as follows:

```
<|im_start|>system You are a smart navigation assistant.<|im_end|>
<|im_start|> Your goal is to find the {object_name}.  What is the next
action?  <|im_end|>
```

The learning rate for the LLM-based model is set to 0.0001. `bfloat16` precision is adopted for efficient inference and reduced memory usage in the LLM-based navigation models. The weight decay is set to 0.1. Additionally, several special tokens are defined. Specifically, the action tokens include `<STOP>`, `<MOVE_FORWARD>`, `<TURN_LEFT>`, `<TURN_RIGHT>`, `<LOOK_UP>`, and `<LOOK_DOWN>`. The observation token, `<|observation|>`, is used to represent the environment's state in the model's input. All other hyperparameters are kept consistent with those used in the BEV-based, transformer-based, and RNN-based models. For the representation consistency loss in our Dual-Path Anti-Forgetting mechanism, we use the $\ell_2$ distance between old and current features on HM3D, and the squared $\ell_2$ distance on MP3D.

## B.2   Details about the Compared Continual Learning Methods

- **LoRA**: Uses a low-rank decomposition with dimension $R = 16$, a scaling factor of $2 \times R$, and a dropout rate of 0.1.
- **LwF**: Applies a KL divergence loss between the current and previous model logits with a coefficient of 0.2.
- **Model Merge**: Linearly interpolates the weights of the current and previous models using a 0.7/0.3 ratio.
- **Data Replay**: Stores $p = 80$ original trajectories per object category for experience replay.
- **Proposed C-Nav**: Stores $p = 80$ trajectory features and samples 50% of each trajectory for adaptive experience selection, ensuring a fair comparison with data replay and uniform sampling methods.

## B.3   The Pseudocode for C-Nav

The pseudocode for C-Nav is illustrated in Algorithm 1. During the initial stage, the model is trained on trajectories from the base object navigation task. Adaptive experience selection is then applied to identify important experiences, which are stored in the feature buffer $\mathcal{B}$. In subsequent learning stages, the model weights from the previous phase are loaded, and the model is optimized by combining

the current task's behavior cloning loss with feature space distillation and feature replay losses. This approach ensures the consistency of the multimodal feature encoder and action decoder across different tasks.

---

**Algorithm 1** C-Nav for Continual ObjectNav with Adaptive Sampling

---

1: **Initialize:** Multimodal encoder $f_0$, decoder $\pi_0$, feature buffer $\mathcal{B} \leftarrow \emptyset$
2: **for** task $k = 1$ to $K$ **do**
3:     **Input:** Dataset $\mathcal{D}_k = \{(s_i^k, c_i^k, \tau_i^k)\}_{i=1}^{N_k}$
4:     **if** $k = 1$ **then**
5:         # perform behavior cloning for initial task
6:         Compute $\mathcal{L}_{\text{Curr}}$
7:         Update $f_k, \pi_k$ by minimizing $\mathcal{L}_{\text{Curr}}$
8:     **else**
9:         # perform feature distillation
10:        Compute $\mathcal{L}_{\text{KD}}$ (Eq. 1) between $f_{k-1}$ and $f_k$
11:        # perform feature replay
12:        Compute $\mathcal{L}_{\text{FR}}$ with $\mathcal{B}$ (Eq. 2)
13:        Update $f_k, \pi_k$ by minimizing $\mathcal{L} = \mathcal{L}_{\text{Curr}} + \lambda_{\text{KD}}\mathcal{L}_{\text{KD}} + \lambda_{\text{FR}}\mathcal{L}_{\text{FR}}$ (Eq. 10)
14:     **end if**
15:     # perform adaptive experience selection in deep feature space
16:     Randomly sample $p$ trajectories from $\mathcal{D}_k$: $\mathcal{T}_p = \{\tau_1^k, \ldots, \tau_p^k\}$
17:     **for** each trajectory $\tau$ in $\mathcal{T}_p$ **do**
18:         Encode frames $\{v_t\}$ via CLIP
19:         Compute LOF scores (Eq. 3–5)
20:         Select keyframes $\mathcal{I} = \{t \mid \text{LOF}_k(v_t) > 1\}$
21:         **for** each $t \in \mathcal{I}$ **do**
22:             Store $(f_k(o_t), a_t)$ in buffer $\mathcal{B}$
23:         **end for**
24:     **end for**
25: **end for**

---

# C   More Results

## C.1   Old vs. New Task Performance per Stage

To provide a more intuitive understanding of how the model adapts to new tasks and the extent of forgetting over time, we report results on the HM3D dataset by separately evaluating performance on new tasks (i.e., target objects in the current stage) and old tasks (i.e., all previously seen target objects, averaged) in Table 6 and Table 7. These results reveal two key trends. First, Finetuning suffers from severe catastrophic forgetting. Although it achieves reasonable performance on new tasks (e.g., 64.63% in Stage 4), its success rate on old tasks drops below 10% after Stage 2. Second, C-Nav achieves a superior trade-off between stability and plasticity. While its new-task performance is slightly lower than that of Data Replay in later stages, it substantially outperforms all baselines on old tasks. Notably, in Stage 4, C-Nav attains an old-task success rate of 42.61%, which is 9.7 percentage points higher than the best baseline (Data Replay at 32.9%). This demonstrates that C-Nav effectively mitigates catastrophic forgetting while preserving the ability to learn new navigation tasks.

Table 6: SR on *new tasks* across stages (HM3D).     Table 7: SR on *old tasks* across stages (HM3D).

| Method | Stage 1 | Stage 2 | Stage 3 | Stage 4 | Method | Stage 1 | Stage 2 | Stage 3 | Stage 4 |
|---|---|---|---|---|---|---|---|---|---|
| Finetuning | 69.63 | 60.55 | 29.54 | 64.63 | Finetuning | 69.63 | 7.72 | 9.83 | 9.05 |
| LoRA | 69.63 | 60.55 | 38.08 | 69.15 | LoRA | 69.63 | 17.99 | 8.19 | 11.70 |
| LwF | 69.63 | 54.27 | 19.57 | 57.98 | LwF | 69.63 | 16.19 | 8.64 | 10.22 |
| Merge | 69.63 | 16.08 | 4.63 | 21.81 | Merge | 69.63 | 66.03 | 41.62 | 25.12 |
| Data Replay | 69.63 | 59.80 | 29.18 | 68.35 | Data Replay | 69.63 | 54.60 | 48.92 | 32.94 |
| C-Nav | 69.63 | 54.52 | 26.33 | 63.30 | C-Nav | 69.63 | 61.38 | 52.27 | 42.61 |

## C.2 Additional Analysis on Feature Consistency Loss

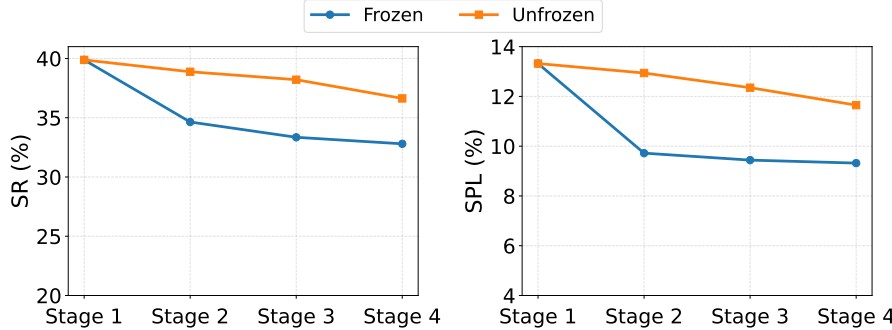

Figure 11: Ablation study: Freezing vs. unfreezing the multimodal encoder on MP3D Continual ObjectNav. We conduct experiments using a transformer-based architecture.

To validate this design choice, we conducted an ablation study on the MP3D Continual ObjectNav dataset by fully freezing the entire multimodal encoder. The results showed a performance drop across learning stages, confirming that full freezing harms generalization and underscoring the necessity of our consistency loss. Our feature consistency loss is designed to learn a shared feature space that remains effective across both past and new tasks. Simply freezing the entire multimodal encoder 11 may preserve old knowledge but limits the model's adaptability, especially when the original feature space does not sufficiently capture new task-specific semantics. Instead, we selectively fine-tune the modality-specific projectors within the encoder while applying consistency regularization to softly align the old and new feature spaces. This design allows the representations to stay connected, yet remain flexible enough to adapt to new tasks. It strikes a balance between preserving past knowledge and enabling plasticity for future learning.

## C.3 Ablation on Loss Weighting Coefficient $\lambda$ (MP3D)

Table 8: Ablation on $\lambda$ for loss weighting (MP3D).

| $\lambda$ | Stage 1 | | Stage 2 | | Stage 3 | | Stage 4 | |
|---|---|---|---|---|---|---|---|---|
| | SR | SPL | SR | SPL | SR | SPL | SR | SPL |
| 1 | 39.88 | 13.32 | 38.00 | 12.25 | 35.41 | 11.00 | 36.22 | 10.85 |
| 5 | 39.88 | 13.32 | 38.37 | 12.29 | 37.32 | 10.97 | 36.81 | 11.90 |
| 10 | 39.88 | 13.32 | 36.66 | 11.80 | 36.68 | 11.45 | 25.29 | 9.51 |

We study the sensitivity of C-Nav to the loss weighting coefficient $\lambda$, which balances the replay loss (for retaining old knowledge) and the current-task loss (for learning new tasks). Table 8 reports success rate (SR) and SPL across four sequential training stages on MP3D, using $\lambda \in \{1, 5, 10\}$; all other model configurations are identical. As $\lambda$ increases, the influence of replayed data as a regularizer grows stronger in later stages. While this helps preserve prior knowledge, an excessively large value (e.g., $\lambda = 10$) over-constrains learning and hinders adaptation to new tasks, resulting in a sharp performance drop in Stage 4 (SR = 25.29%). On the other hand, $\lambda = 1$ offers too little regularization, causing gradual forgetting of previously acquired skills. The setting $\lambda = 5$ strikes an effective balance: it yields slightly better overall performance than $\lambda = 1$ while maintaining strong retention of old tasks. Consequently, $\lambda = 5$ achieves the best trade-off between stability and plasticity and is adopted in all main experiments.

## C.4 Results for the Dual-Path Anti-Forgetting Performance at Each Stage

To intuitively analyze the contribution of each component in C-Nav, we visualize the SR and SPL curves for each stage. As shown in the Figure 12, removing any component leads to a significant drop in both SR and SPL across all four model architectures and two datasets. Additionally, we report the average success rate for each component across the four architectures in Figure 13. The results

reveal that the multimodal encoder is prone to bias across tasks, leading to more severe forgetting. Specifically, removing the KD component causes substantial performance degradation: SR drops by 22.15 and 15.92 on HM3D-SR-Avg and MP3D-SR-Avg, respectively, and by 24.15 and 16.50 on HM3D-SR-Last and MP3D-SR-Last.

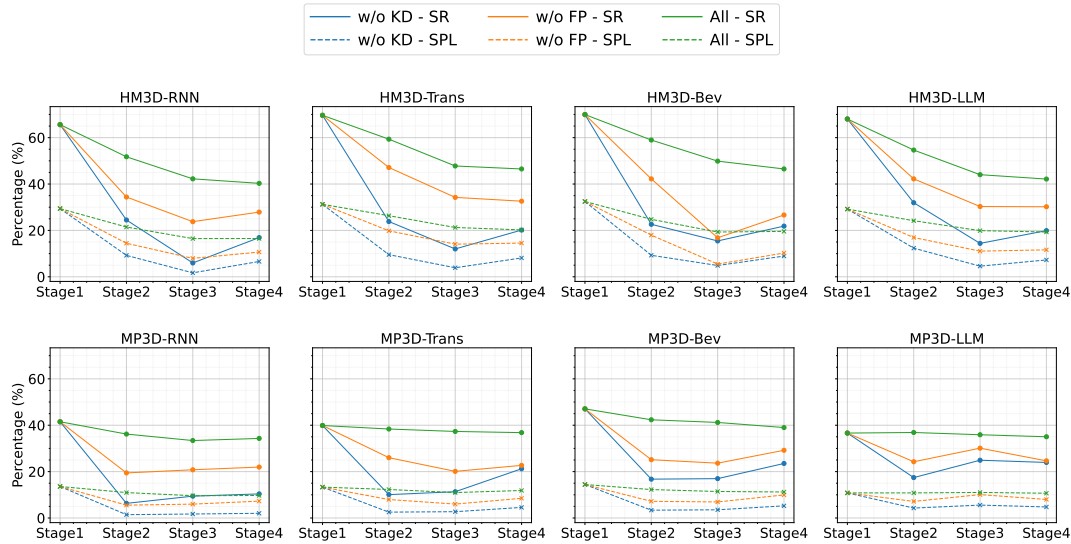

Figure 12: Results for the dual-path anti-forgetting performance at each stage.

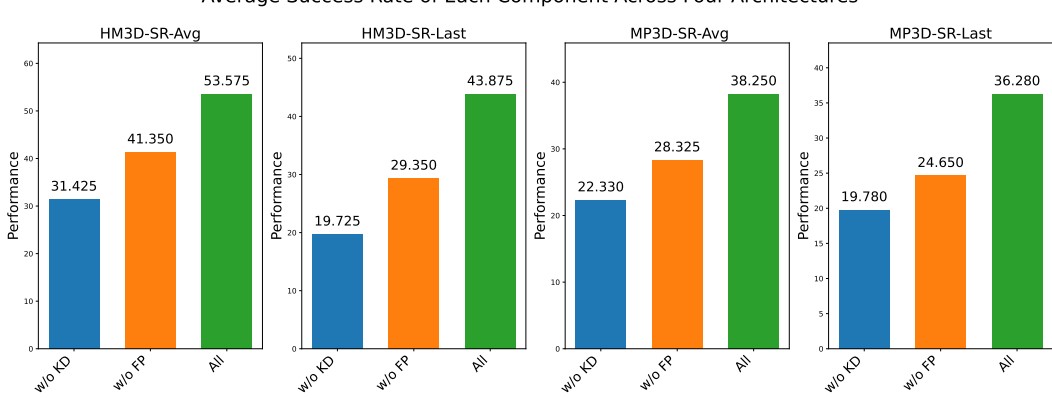

Figure 13: Average success rate for each component across four architectures. HM3D-SR-Avg and MP3D-SR-Avg denote the mean SR over all stages, while HM3D-SR-Last and MP3D-SR-Last indicate the mean SR in the final stage, averaged across architectures.

## C.5    Results for the Adaptive Experience Selection at Each Stage

In Figure 14, we visualize the SR and SPL curves across different training stages under various sampling strategies. Our adaptive sampling method achieves performance comparable to training without any sampling, while clearly outperforming both uniform interval sampling and naive data replay approaches. As shown in Figure 15, the gains are particularly evident when compared to data replay: our method improves the average success rate by 0.8 and 2.0 on HM3D-SR-Avg and MP3D-SR-Avg, respectively, and by 0.625 and 4.625 on HM3D-SR-Last and MP3D-SR-Last. Compared to uniform sampling, our approach yields even larger improvements, boosting HM3D-SR-Last and MP3D-SR-Last by 3.375 and 7.3, respectively. These results demonstrate that our adaptive sampling strategy not only maintains competitive overall performance but also significantly enhances learning stability and final-stage effectiveness. By selective revisitation of information-rich trajectory frame features, the model better retains knowledge across tasks, leading to more robust continual learning in navigation scenarios

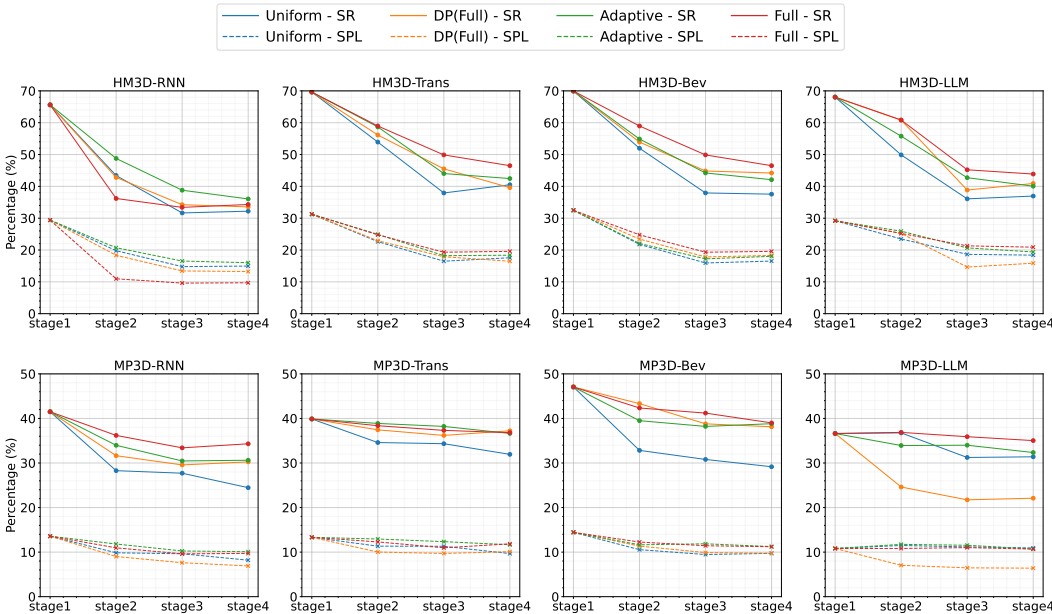

Figure 14: Results for the adaptive experience selection at each stage.

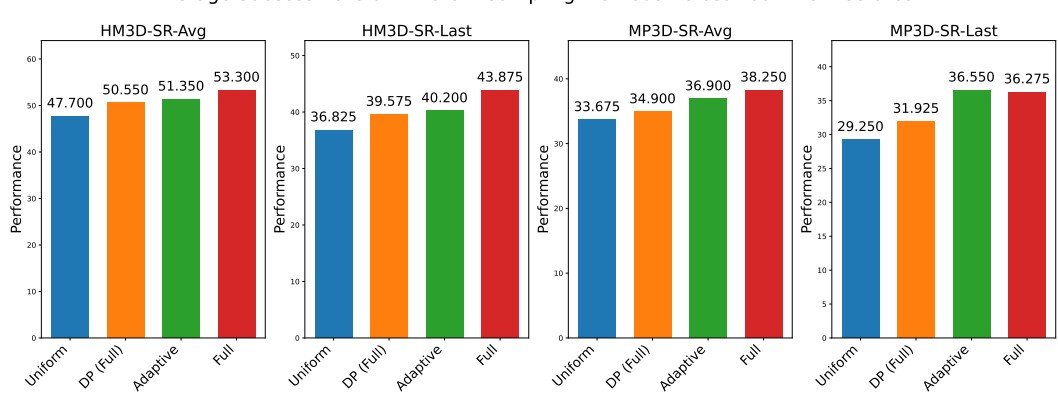

Figure 15: Average success rate for different sample methods across four architectures.

