# OpenReview forum: "C-NAV: Towards Self-Evolving Continual Object Navigation in Open World"
_NeurIPS.cc/2025/Conference — NeurIPS 2025 poster_

### Official Review · Reviewer_23E4 · 2025-07-03

**Clarity:** 3
**Significance:** 2
**Originality:** 2
**Rating:** 3
**Confidence:** 4

**Summary:**

This submission introduces a continual object navigation benchmark based on the HM3D and MP3D datasets, where agents are required to incrementally learn new object categories while preventing catastrophic forgetting. The proposed C-Nav framework mitigates forgetting through a dual-path mechanism that combines feature-space distillation to preserve representation consistency and feature replay to stabilize policy learning. Additionally, an adaptive experience selection strategy based on the Local Outlier Factor (LOF) reduces storage overhead. The framework is evaluated across four navigation architectures using SR and SPL metrics under staged learning settings.

**Questions:**

- Would it be possible to report separate performance on old and new tasks at each stage? This might provide a more intuitive understanding of how the model adapts to new tasks and the extent of forgetting over time.

- Why is the L2 distance constraint between features from different stages effective in preventing representation drift? Is there any theoretical or empirical justification supporting this design choice?

- One observation is that while several baselines exhibit performance recovery from stage 3 to stage 4, C-Nav continues to decline. What factors might account for this trend, and does it suggest potential limitations in the current mitigation mechanism under longer task sequences?

**Ethical Concerns:**

["NO or VERY MINOR ethics concerns only"]

**Final Justification:**

After rebuttal, I believe some of my concerns remain unaddressed.

**Limitations:**

The author discussed the limitations on page 9.

**Paper Formatting Concerns:**

No major formatting concerns were identified.

**Quality:**

3

**Strengths And Weaknesses:**

## Strengths
- The overall presentation of the work is clean and easy to follow.

- The submission establishes a clear continual learning task setting and benchmark for object navigation, which provides a useful foundation for future research in this area.

- The evaluations are thorough, with experiments across diverse architectures (Transformer, RNN, BEV, LLM), and the visualizations help illustrate performance differences across stages.

## Weakness
- It is unclear whether there is a real need for continual learning in a task with only 6 categories, such as HM3D. The necessity of constructing a staged continual learning setting under such a limited scenario may lack a clear correspondence to real-world applications.

- There is a lack of discussion or justification for key hyperparameters, such as the loss weighting coefficients and the threshold used in the LOF-based frame selection.

- The comparison is limited to classical continual learning methods, and does not include more recent or advanced approaches.

- Qualitative trajectory analysis is absent, which limits intuitive understanding of behavior consistency and the retention of task-specific strategies across continual stages.

---

> ### Author Rebuttal · Authors · 2025-07-30
>
> We greatly appreciate your insightful comments and suggestions, as they have been helpful in refining and enhancing our work. We have thoroughly reviewed all of your points and have addressed your concerns as outlined below:
>
>
> > **Q1: Clarification on the Need for Continual Learning with Limited Object Categories**
>
> Thank you for raising this important question. While HM3D contains only 6 categories, we also evaluate on MP3D, which offers **21 diverse and semantically rich categories** (e.g., gym_equipment, fireplace), providing a more realistic benchmark.
>
> Our goal is not merely to increase category diversity, but to **address fundamental challenges in continual learning for embodied object navigation**—a setting largely overlooked in prior work. Existing methods are primarily designed for static perception tasks like image classification or retrieval. In contrast, our task involves **long-horizon exploration, multimodal input, obstacle avoidance, and active perception**, introducing unique and complex forgetting patterns.
>
> Even under controlled staged settings, we observe **catastrophic forgetting, semantic confusion, and task interference**, which significantly degrade navigation performance and are **not effectively addressed by current methods**.
>
> Our framework offers a scalable and diagnosable platform to study these challenges. We hope this work provides valuable insights toward building embodied agents that are adaptive, efficient, and capable of long-term deployment in dynamic environments.
> > **Q2: Ablation Studies**
>
> In response, we now provide detailed ablation studies on key hyperparameters.
>
> (1) Loss Weighting Coefficients (MP3D)
>
>
> |λ|Stage1| |Stage2| |Stage3| |Stage4| |
> |-|-|-|-|-|-|-|-|-|
> | |SR|SPL|SR|SPL|SR|SPL|SR|SPL|
> |1|39.88|13.32|38|12.25|35.41|**11**|36.22|10.85|
> |5|39.88|13.32|**38.37**|**12.29**|**37.32**|10.97|**36.81**|**11.9**|
> |10|39.88|13.32|36.66|11.8|36.68|11.45|25.29|9.51|
>
> As shown in the table,  we chose λ = 5 as the optimal weight for both losses, **balancing the retention of old knowledge and the ability to learn new tasks effectively.**
>
>
> (2) LOF Frame Retention Ratio (MP3D)
>
> |Retention Ratio|Stage1| |Stage2| |Stage3| |Stage4| |
> |-|-|-|-|-|-|-|-|-|
> | |SR|SPL|SR|SPL|SR|SPL|SR|SPL|
> |0.1|39.88|13.32|33.45|11.53|30.8|9.72|34.17|10.84|
> |0.3|39.88|13.32|37.9|12.14|37.47|12.07|36.17|11.61|
> |0.5|39.88|13.32|**38.88**|**12.94**|**38.21**|**12.35**|36.63|11.65|
> |1|39.88|13.32|38.37|12.29|37.32|10.97|**36.81**|**11.86**|
>
> As shown in the table, the model performance decreases as the retention ratio drops. However, the overall decline is relatively mild. Notably, **even when retaining only 10%** of the key frames, our method still demonstrates competitive performance, highlighting the robustness of our approach in selecting relevant experiences for continual learning.
> > **Q3: Include More Recent Continual Learning Methods**
>
> Thank you very much for your valuable feedback. In our experiments, we compared C-Nav with several representative continual learning baselines, including Data Replay, LoRA, LwF, and Model Merge. Among these, Data Replay serves **as a strong baseline** due to its ability to mitigate catastrophic forgetting.
>
> Our results demonstrate that the proposed C-Nav outperforms these methods not only in terms of navigation performance, but also **offers clear advantages in storage efficiency and data privacy**, thanks to our adaptive experience selection and dual-path forgetting mitigation strategy.
>
> We have also added a comparison with **ZSCL**[1], a recent method that imposes constraints in both parameter and feature space to support continual learning. The results are summarized below:
>
> |Methods(HM3D)|Stage1|Stage2|Stage3|Stage4|
> |---|---|---|---|---|
> | |SR|SR|SR|SR|
> |Finetuning|69.63|23.38|13.24|19.5|
> |Lora|69.63|30.6|13.36|22.5|
> |LWF|69.63|27.48|10.53|19.2|
> |MAE|69.63|51.23|35.22|24.5|
> |Data Replay|69.63|56.14|45.5|39.6|
> |**ZSCL**|69.63|31.72|25.55|24.6|
> |Ours|69.63|**59.34**|**47.78**|**46.5**|
>
> While ZSCL performs reasonably well in early stages, its performance declines significantly in later stages. For example, in Stage 4, its SR **drops to 24.6, compared to 46.5 achieved by C-Nav**. This reflects a broader limitation of many continual learning methods that are primarily developed for **static, perception-focused tasks such as image classification or image-text matchin**g. These tasks do not involve **long-horizon exploration, active perception, or obstacle avoidance**, which are essential for embodied navigation.
>
> In contrast, our task setting integrates these embodied challenges, and C-Nav is specifically designed to address them, enabling more effective continual learning in complex, dynamic environments.
>
> We are further expanding our comparisons to include additional recent methods, and the updated results will be presented in the final version of the paper.
>
> We hope this response sufficiently addresses your concerns.
>
> [1]Preventing Zero-Shot Transfer Degradation in Continual Learning of Vision-Language Models
> > **Q4: Include Qualitative Trajectory Analysis**
>
> Thank you for the suggestion. We provide qualitative trajectory analysis in **Section C.3 (Case Study) of the supplementary material**, including:
> - **Four TopDown visualizations** (Figure 7 & 8), illustrating agent behavior across stages;
> - **Four video clips**, showing first-person navigation trajectories of both the baseline model and C-Nav when evaluated on old tasks after multi-stage training.
>
> During evaluation, we focused on two episodes: locating a **toilet (stage 1)** and **a bed (stage 2)**. The visualizations in Figure 7 and Figure 8 clearly show that the **baseline agent exhibits catastrophic forgetting**. After multi-stage training, it consistently navigates toward the sofa (the target in the final stage), completely forgetting the navigation strategy for older objects. In contrast, the C-Nav model **successfully navigates to both the toilet and bed targets**, demonstrating superior long-term task retention.
> To provide a more intuitive comparison, we plan to **move these trajectory visualizations to the main experimental results section** in future versions of the paper. This will better highlight the challenges in continual object navigation and allow a clearer comparison of model performance.
> > **Q5: Report Old vs. New Task Performance per Stage**
>
> Following your suggestion, we report the results on the HM3D dataset by separately evaluating performance on old and new tasks.
>
> New Task Performance:
> |Methods|Stage1|Stage2|Stage3|Stage4|
> |-|-|-|-|-|
> |Finetuning|69.63|60.55|29.54|64.63|
> |LoRA|69.63|60.55|38.08|69.15|
> |Lwf|69.63|54.27|19.57|57.98|
> |Merge|69.63|16.08|4.63|21.81|
> |Data Replay|69.63|59.80|29.18|68.35|
> |C-Nav|69.63|54.52|26.33|63.30|
>
> Old Task Performance:
> |Methods|Stage1|Stage2|Stage3|Stage4|
> |-|-|-|-|-|
> |Finetuning|69.63|7.72|9.83|9.05|
> |LoRA|69.63|17.99|8.19|11.70|
> |Lwf|69.63|16.19|8.64|10.22|
> |Merge|69.63|66.03|41.62|25.12|
> |Data Replay|69.63|54.60|48.92|32.94|
> |C-Nav|**69.63**|**61.38**|**52.27**|**42.61**|
>
> These results clearly demonstrate that:
> - **Finetuning** suffers from severe **catastrophic forgetting**, with old-task performance dropping drastically as new tasks are introduced.
> - **C-Nav** maintains stable performance across both old and new tasks, effectively balancing stability and plasticity. In the final stage, it achieves a **10% higher success rate** on old tasks compared to the best-performing baseline.
>
> We will include this breakdown in **Appendix C.4**, along with further analysis to highlight the trade-offs involved in continual object navigation.
> >**Q6: Justify L2 Feature Constraint for Representation Stability**
>
> We adopt an L2-based feature distillation loss to enforce consistency in the encoder's representations across stages. This simple yet effective constraint mitigates representation drift.
> The design is inspired by **prior work on feature-level knowledge distillation** [2], where **L2 regularization helps preserve essential semantic structures from earlier tasks**. Ablation results (Table 4) show that removing this constraint leads to substantial performance degradation (**−22.15 SR on HM3D and −15.92 on MP3D**). This confirms that stabilizing the encoder’s feature space is crucial for long-term retention, and supports the effectiveness of our L2 constraint in mitigating forgetting.
>
> [2] CVPR 2019：Relational knowledge distillation.
> >  **Q7: C-Nav Shows Performance Decline in Stage 4 While Several Baselines Recover**
>
> As shown in **Answer 2**, all baseline methods experience significant performance drops on old tasks, indicating **severe forgetting**. In contrast, C-Nav effectively mitigates this, **achieving a 10% higher success rate** on old tasks than the best baseline in the final stage.
>
> The recovery in some baselines during Stage 4 is likely due to two factors:
> - **Task Difficulty**: Stage 4 introduces "sofa" as the target, which is easier than "TV" in Stage 3. The base model's new-task success rate jumps from 29.54% in Stage 3 to 64.63% in Stage 4, suggesting **overfitting to the easier task, rather than true continual learning**.
> - **Semantic Similarity**: The target "sofa" is visually similar to previously learned objects like "bed" and "chair," leading to potential confusion and misidentification, which could **falsely improve performance on old tasks**.
>
> In contrast, C-Nav prioritizes consistent performance across stages, focusing on **long-term knowledge retention** over short-term gains. This slight decline in Stage 4 reflects its cautious adaptation strategy, designed to prevent overfitting and ensure stability across extended task sequences.
>
>
> Thank you for your valuable feedback. We hope our responses address your concerns and are happy to clarify further if needed.

---

> > ### Author Response · Authors · 2025-08-06
> > **Looking forward to your feedback**
> >
> > Dear reviewer 23E4:
> >
> > We sincerely appreciate your insightful and thoughtful comments!  Following your suggestions, we have provided detailed explanations, added visualizations, and included additional experimental results.
> > Specifically, we moved the visualizations from the supplementary materials into the main text, provided ablation studies on key hyperparameters, expanded our comparisons with additional baselines, and reported both the old and new task performance at each stage to better present the results.
> >
> > As it is approaching the end of the discussion period (Aug 8 AoE), we sincerely hope you could look through our response and have a further comment at your convenience if you have any questions about the paper. We will do our best to address the issues of the reviewer.
> >
> >
> > Best wishes,
> >
> > Submission 22093 Authors.

---

> ### Author Response · Authors · 2025-08-08
> **Follow-up on Rebuttal for Submission 22093**
>
> Dear Reviewer 23E4,
>
> As the discussion phase is closing soon (Aug 8 AoE), we sincerely hope you may have a chance to revisit our rebuttal and share any further comments or questions. Your feedback has been invaluable, and we would greatly appreciate any final suggestions.
>
> We would also like to briefly reiterate the main contributions of our paper, highlighting how our approach differs from prior work:
>
> - **New Benchmark:** We introduce the first benchmark for continual object navigation, advancing beyond static tasks like image classification or vision-language retrieval to an embodied setting that demands long-horizon exploration, active perception, obstacle avoidance, spatial reasoning, and multi-modal integration. These challenges introduce complex forgetting dynamics not seen in traditional benchmarks.
>
> - **C-Nav Framewo**rk: We propose a continual object navigation framework that combines feature distillation to maintain representation consistency and feature replay to ensure decision-making consistency throughout training. This approach addresses both encoder drift and policy degradation across sequential task learning.
>
> - **Adaptive Experience Selection**: We develop a memory-efficient sampling strategy that leverages Local Outlier Factor (LOF) in the feature space to identify semantically meaningful frames. This method retains key experiences while significantly reducing storage requirements and mitigating privacy risks related to raw trajectory retention.
>
> - **Strong Results**: We demonstrate that C-Nav consistently outperforms strong baselines across four distinct navigation architectures, achieving superior memory efficiency and retention of prior knowledge compared to full replay methods.
>
> In response to your concerns, we provide the following point-by-point clarifications:
>
> - **Visualization**: Our supplementary material already contains qualitative trajectory visualizations and video clips showing that baseline models suffer catastrophic forgetting, while C-Nav preserves navigation abilities across multiple stages. We have now moved these visualizations into the main experimental results section and added a subfigure in the introduction to better illustrate the continual object navigation task and the forgetting challenge.
>
> - **More Results**: We now include additional ablation studies on loss weighting coefficients and LOF frame retention ratios, further demonstrating the robustness of our method even with minimal frame retention. Additionally, we have added new results comparing old vs. new task performance across stages, as well as a comparison with the newly introduced baseline method, ZWCL.
>
> - **Clarifications**: We have clarified the necessity of our benchmark for embodied object navigation tasks and discussed the performance comparison between C-Nav and baselines in the context of multi-stage continual learning.
>
> We are happy to provide further clarification if needed. Thank you once again for your time and constructive review.
>
> Best regards,
>
> Submission 22093 Authors

---

> > ### Comment · Reviewer_23E4 · 2025-08-08
> > **Post-Rebuttal Comments**
> >
> > I thank the authors for their rebuttal, which has addressed some of my initial questions. However, I still have the following concerns:
> >
> > - Comparison with more recent or advanced approaches – My original request for “comparison with more recent or advanced approaches” was specifically aimed at seeing results against existing zero-shot ObjectNav methods, particularly map-based ones. Current zero-shot ObjectNav approaches already achieve strong performance; for example, UniGoal reaches 41.0 SR and 16.4 SPL on MP3D. In contrast, the proposed method requires end-to-end training and still underperforms, which raises doubts about its necessity.
> >
> > - Advantage of map-based methods – Continuing from point 1, for the Continual-ObjectNav problem, I believe map-based methods have an inherent advantage over end-to-end methods: as more observations are gathered, the map becomes more complete, making the environment fully known. At that point, navigation essentially reduces to a point-to-point problem.
> >
> > - Similarity to existing benchmarks – The proposed Continual-ObjectNav benchmark appears quite similar to the existing MultiON benchmark, which makes the novelty less clear.
> >
> >
> > Overall, my main concerns have not been sufficiently resolved, and I will keep my rating for reject.

---

> ### Author Response · Authors · 2025-08-08
>
> Thank you for your comments. We would like to clarify the following point:
>
> > **Q1: Difference between Continual-ObjectNav and MultiON tasks**
>
> While MultiON also involves multiple objects, it is **fundamentally different** from our benchmark:
>
> - **MultiON**: Uses **a single-stage training setup**, where **all object categories are available during the single training phase**. At test time, the agent sequentially searches for multiple targets **within the same environment**, evaluating its **long-term spatial memory**. Completing Subtask 1 can facilitate the search for Subtask 2 (e.g., the object for Subtask 2 may have already been observed while searching for Subtask 1).
>
> - **Continual-ObjectNav**: Uses **a multi-stage training process**, where **trajectories for new categories are introduced over time**, **without access to all earlier training data**. In each stage, the agent **learns to navigate to new object categories**, and at test time, its navigation performance is evaluated in **unseen environments for all previously learned categories**. This setup falls under multi-stage incremental learning, introducing challenges such as **catastrophic forgetting** and the **stability–plasticity trade-off**, which do not occur in MultiON.
>
> Therefore, our benchmark not only measures navigation performance but also **emphasizes the agent’s ability to retain knowledge after multi-stage training**.
>
>
> > **Q2: Comparison with Recent Zero-shot Methods**
>
> We appreciate the reviewer’s observation that current map-based Zero-shot ObjectNav methods, **such as UniGoal**, have made significant progress. However, **End-to-End (E2E) methods have unique advantages** in the following aspects:
>
> - **Deployment and Efficiency**: Zero-shot methods heavily rely on large vision-language models (e.g., GPT-4o) and multiple hand-crafted rules (e.g., object detection confidence thresholds, subgraph matching thresholds, passable area determination), which results in high deployment costs and latency. In contrast, E2E models have a **more straightforward architecture (without requiring specialized hyperparameters) and offer faster inference, making them more suitable for resource-constrained navigation systems**.
>
> - **Performance Advantage**: When **complete training data** for all object categories is available, **E2E methods can outperform Zero-shot methods**. For instance, our baseline BEV-based end-to-end model [1] achieves a success rate (SR) of 65.2% and 47.74% on HM3D and MP3D, respectively, while UniGoal reports 54.5% and 41.0%, **leading to improvements of approximately 10.7% and 6.7%.** Furthermore, incorporating auxiliary losses and reinforcement learning fine-tuning [3] would **further enhance performance.**
>
>     |Model|SR（HM3D）|SPL（HM3D）|SR（MP3D）|SPL（MP3D）|
>     |---|---|---|---|---|
>     |Uni-Goal[2]|54.5|25.1|41.0|**16.4**|
>     |BEV-Based[1]|**65.2**|**29.56**|**47.74**|15.12|
>
> **Research Motivation**: Despite the superior performance under sufficient data, **E2E methods** suffer from significant **catastrophic forgetting** in Continual ObjectNav tasks. This is the main motivation for introducing the Continual-ObjectNav task and methodology. Our goal is to enable the model to continuously learn navigation for new categories while avoiding catastrophic forgetting.
>
> - The agent must gradually **learn to navigate new categories** without revisiting all previously seen training data.
> - **Accumulating knowledge across multiple training stages is crucial**, with the avoidance of catastrophic forgetting being the key challenge.
>
> [1] Object Goal Navigation with Recursive Implicit Maps.
>
> [2] UniGoal: Towards Universal Zero-shot Goal-oriented Navigation.
>
> [3] PIRLNav: Pretraining with Imitation and RL Finetuning for ObjectNav.
> > **Q3: Regarding Map-Based Methods**
>
> The advantages of **map-based methods** mentioned by the reviewer are primarily applicable **under the Multi-ON task setting**. In Multi-ON, the agent searches for multiple targets sequentially within **the same environment**, allowing it to **continuously build and leverage a global map, which in turn accelerates the exploration of subsequent targets**.
>
> However, in the **Continual-ObjectNav task**, the focus is on evaluating the agent’s semantic navigation ability in **unseen environments across multiple learning stages**. Each test episode is conducted in **a previously unseen environment and involves navigating to a single target (which may belong to either the current or a previously learned category)**. As such, there is **no presence of multiple subtasks within the same environment**, and thus map-based methods **cannot exploit cross-subtask accumulation of spatial information or maps** during testing.
>
> Because each test occurs in a **novel environment with a single target**, the agent cannot build persistent maps through repeated exploration. Instead, it must rely on **semantic-visual knowledge learned across training stages** to perform the search effectively.

---

> > ### Comment · Reviewer_23E4 · 2025-08-09
> > **Comments**
> >
> > Thanks the authors‘ detailed rebuttal. Unfortunately, I believe some of my concerns remain unaddressed. The motivation for Continual-ObjectNav is framed around catastrophic forgetting and the stability–plasticity trade-off. However, these issues predominantly affect end-to-end ObjectNav methods, while map-based methods are generally not subject to the same challenges. As a result, the proposed method appears applicable only to a subset of ObjectNav methods.
> >
> > My main concern is rooted in the current technical trend of the ObjectNav field. Zero-shot ObjectNav—particularly map-based zero-shot methods—has been gaining momentum and is becoming a dominant paradigm. These methods require no training in the target environment and inherently avoid catastrophic forgetting issues. Given this trend, I am doubtful that the proposed approach will contribute meaningfully to the current methods.
> >
> > Furthermore, in response to Q2, the reported MP3D results do not surpass zero-shot baselines. The reply to Q3 also does not align well with my concern: current zero-shot ObjectNav methods evaluate in unseen environments, which differs from the same environment setting described in the rebuttal.
> >
> > Regrettably, my original concern still stands.

---

> ### Author Response · Authors · 2025-08-09
>
> > **Concerns Regarding End-to-End Methods**
>
> We acknowledge the reviewer’s point that current Zero-shot methods have made substantial progress and offer strong generalization capabilities. However, **End-to-End (E2E) models remain an important research direction in navigation tasks** [1][3][4][5], as they **provide clear advantages in inference speed, on-device deployment, and performance**.
>
>
> We have added the results of the BEV-based baseline model with auxiliary loss [1] in the Table. This model achieves an SR of 50.25 and SPL of 17.00 on MP3D, **representing improvements of 9.25 and 0.6 over UniGoal on the same dataset**. In addition, recent works such as UniNavid [4] and TrackVLA [5] also demonstrate the performance advantages of E2E methods. For example,  by incorporating more navigation data [4], it achieves **approximately a 20% SR improvement** over UniGoal on the HM3D dataset. Furthermore, E2E methods can make effective use of newly collected data for model fine-tuning, which is something that Zero-shot methods cannot easily achieve.
>
>
> |Model|SR（HM3D）|SPL（HM3D）|SR（MP3D）|SPL（MP3D）|
> |---|---|---|---|---|
> |Uni-Goal[2]|54.5|25.1|41.0|**16.4**|
> |BEV-Based[1]|65.2|29.56|47.74|15.12|
> |BEV-Based[1] （with Aux Loss）|-|-|**50.25**|**17.00**|
> |UniNavid[4]|**73.7**|**37.1**|-|-|
>
>
> While **E2E methods have these advantages**, they **suffer from catastrophic forgetting** in the Continual Object Navigation setting. This is **the main reason our work focuses on E2E methods for Continual Object Navigation**, aiming to address the challenge of retaining and accumulating knowledge and skills across multiple learning stages.
>
> In existing Zero-shot ObjectNav, **since Zero-shot methods do not involve any training, they avoid forgetting caused by parameter updates**. However, these methods **struggle to fine-tune models using newly collected navigation data, which can result in suboptimal performance in some cases**.
>
> In contrast, **our benchmark focuses on how well a model retains prior knowledge after multiple training stages that use new navigation trajectories, enabling an embodied agent to continually adapt to new instructions**. Both training and evaluation in our benchmark are conducted in a multi-phase, incremental manner.
>
> As the importance of Zero-shot methods, **we will add a discussion on the role of Zero-shot methods, such as UniGoal, in Continual ObjectNav in the related work section.**
>
>
> [1] Object Goal Navigation with Recursive Implicit Maps.
>
> [2] UniGoal: Towards Universal Zero-shot Goal-oriented Navigation.
>
> [3] PIRLNav: Pretraining with Imitation and RL Finetuning for ObjectNav.
>
> [4]Uni-NaVid: A Video-based Vision-Language-Action Model for Unifying Embodied Navigation Tasks
>
> [5]TrackVLA: Embodied Visual Tracking in the Wild
>
> We thank the reviewers again for their valuable feedback and constructive suggestions. We hope our clarifications address the concerns raised, and we are happy to provide further explanation if there are any remaining questions.

---

### Official Review · Reviewer_3LhB · 2025-07-03

**Clarity:** 2
**Significance:** 3
**Originality:** 3
**Rating:** 4
**Confidence:** 3

**Summary:**

This paper addresses the underexplored but important problem of continual object goal navigation, where an embodied agent must incrementally learn to navigate to novel object categories without forgetting previously learned ones. The authors propose a new benchmark split for ObjectNav tasks on HM3D and MP3D, with objects introduced in multiple stages. To mitigate catastrophic forgetting, the authors propose a dual-path anti-forgetting strategy, combining (1) feature distillation to stabilize the visual encoder and (2) feature replay based on latent representations. Additionally, an Adaptive Experience Selection (AES) module selects representative frames for memory storage using semantic outlier detection. This design results in a lightweight, modular solution to continual navigation

**Questions:**

- Have you tried evaluating the agent across different stages on similar or re-occurring objects? For example, if "sofa" appears in stage 1 and again in stage 4 under different visual conditions, how well does the model preserve category identity or avoid overwriting knowledge?

**Ethical Concerns:**

["NO or VERY MINOR ethics concerns only"]

**Final Justification:**

After receiving the criticism, I was able to understand the C-Nav model. The author may need to update the writing.
I recommend boderline accept.

**Limitations:**

- Lack of qualitative analysis and visualization: The paper primarily presents numerical results (SR, SPL) but lacks qualitative analysis such as trajectory visualizations, failure cases, or behavior comparisons. This makes it harder to interpret what the model has learned or how forgetting manifests in practice.

- Insufficient explanation of figures: For example, Figure 1 lacks accompanying explanation or description in the main text, which may confuse readers unfamiliar with the setting. Better figure captions or integration into the discussion would improve clarity.

- No analysis of stage-order sensitivity: The current results are based on one fixed order of object introduction. However, in continual learning, the order of tasks can affect performance and forgetting. Exploring stage permutation effects would improve the robustness claims.

**Paper Formatting Concerns:**

No concerns

**Quality:**

2

**Strengths And Weaknesses:**

### Strengths:
- Interesting motivation: The problem setting is realistic and highly applicable to embodied agents deployed in open-world environments.

- Comprehensive contribution: The paper offers a benchmark design, a novel anti-forgetting method,


### Weakness:
- The method section is dense and may be difficult to follow for readers unfamiliar with continual learning or distillation techniques. More diagrams or intuition would help.

- The paper lacks visualizations or qualitative examples, such as trajectory comparisons which would help readers understand what the model is learning or how retrieval is affecting behavior.

- The continual learning baselines used for comparison are somewhat dated, It would strengthen the evaluation to include more recent rehearsal.

---

> ### Author Rebuttal · Authors · 2025-07-29
>
> We greatly appreciate your insightful comments and suggestions, as they have been helpful in refining and enhancing our work. We have thoroughly reviewed all of your points and have addressed your concerns as outlined below:
> > **Q1: Clarify Method with More Diagrams and Intuition.**
>
> Thank you very much for your feedback. We have revised both the Introduction and Figure 1 accordingly.
>
> - Figure 1 has been updated with **a new subfigure (b)** to visually demonstrate how the agent loses its ability to navigate to previously learned goals after learning new tasks, thus illustrating the challenge of catastrophic forgetting in continual object navigation.
> - We revised the **figure caption** to clarify the key concepts. The new caption reads: Continual Object Navigation: The agent must continually learn from new data while retaining its ability to navigate to previously seen object goals. Subfigure (b) shows representative trajectories from a baseline model and C-Nav when evaluated on an old task after learning a new one.
> - Additionally, we have added **an explanation of Figure 1** in the Introduction, emphasizing the underlying challenges of continual learning. Specifically:
> As the agent is exposed to new tasks over time, distributional shifts in sensory inputs and action patterns introduce biases into the model components, leading to representation drift and policy degradation. This results in catastrophic forgetting of previously acquired knowledge, **as illustrated in Figure 1**. The agent often becomes capable of semantic exploration only for objects in the current task, losing its ability to navigate to goals learned earlier.
>
>
> **The algorithm pseudocode and trajectory visualizations in the supplementary material (already included in the original submission)** provide a more detailed and intuitive understanding of our method.
>
> These updates make the motivation and challenges of continual object navigation more visually accessible and help contextualize our proposed solution early in the paper.
>
>
> > **Q2: Add Visualizations or Qualitative Examples**
>
> Thank you for the suggestion. We provide qualitative trajectory analysis in **Section C.3 (Case Study) of the supplementary material (already included in the original submission)**, including:
> - **Four TopDown visualizations** (Figure 7 & 8), illustrating agent behavior across stages;
> - **Four video clips**, showing first-person navigation trajectories of both the baseline model and C-Nav when evaluated on old tasks after multi-stage training.
>
> During evaluation, we focused on two episodes: locating a **toilet (stage 1)** and **a bed (stage 2)**. The visualizations in Figure 7 and Figure 8 clearly show that the **baseline agent exhibits catastrophic forgetting**. After multi-stage training, it consistently navigates toward the sofa (the target in the final stage), completely forgetting the navigation strategy for older objects. In contrast, the C-Nav model **successfully navigates to both the toilet and bed targets**, demonstrating superior long-term task retention.
>
> To provide a more intuitive comparison, we plan to **move these trajectory visualizations to the main experimental results section** in future versions of the paper. This will better highlight the challenges in continual object navigation and allow a clearer comparison of model performance.
> > **Q3: Include More Recent Rehearsal Baselines**
>
> Thank you very much for your valuable feedback. In our experimental section, we have compared our method against several representative continual learning baselines, including Data Replay, LoRA, LwF, and Model Merge. Among them, Data Replay serves **as a strong baseline** due to its effectiveness in mitigating catastrophic forgetting.
> Our results demonstrate that the proposed C-Nav outperforms these methods not only in terms of navigation performance, but also **offers clear advantages in storage efficiency and data privacy**, thanks to our adaptive experience selection and dual-path forgetting mitigation strategy.
>
>
> We have also added a comparison with **ZSCL**[1], a recent method that imposes constraints in both parameter and feature space to support continual learning. The results are summarized below:
>
> |Methods(HM3D)|Stage1| |Stage2| |Stage3| |Stage4| |
> |---|---|---|---|---|---|---|---|---|
> | |SR|SPL|SR|SPL|SR|SPL|SR|SPL|
> |Finetuning|69.63|31.25|23.38|9.14|13.24|3.57|19.5|7.59|
> |Lora|69.63|31.25|30.6|11.85|13.36|4.95|22.5|9.58|
> |LWF|69.63|31.25|27.48|9.5|10.53|2.55|19.2|6.94|
> |MAE|69.63|31.25|51.23|22|35.22|12.99|24.5|7.19|
> |Replay|69.63|31.25|56.14|22.91|45.5|17.88|39.6|16.48|
> |**ZSCL**|69.63|31.25|31.72|11.3|25.55|9.21|24.6|8.84|
> |Ours|69.63|31.25|**59.34**|**26.35**|**47.78**|**21.26**|**46.5**|**20.22**|
>
> Although ZSCL performs reasonably well in earlier stages, its performance drops considerably in later stages. For example, in Stage 4, **its SR drops to 24.6 compared to 46.5 achieved by C-Nav**.
> This highlights a key limitation of existing continual learning frameworks, which are often designed for **static, perception-focused tasks like image classification or image-text matching**.
> These tasks do not involve **long-horizon exploration, target identification, active perception, or obstacle avoidance**, which are essential components of continual object navigation.
> As a result, such methods are difficult to directly apply to this setting.
> In contrast, our task explicitly incorporates these embodied challenges, and **C-Nav is designed to handle these embodied challenges, enabling effective continual learning in complex, dynamic settings**.
>
> We are also extending our comparisons to include additional recent methods, and the results will be reported in the final version of the paper.
>
> We hope this response sufficiently addresses your concerns.
>
>
> [1]Preventing Zero-Shot Transfer Degradation in Continual Learning of Vision-Language Models
>
> > **Q4: Evaluate Category Identity Preservation Across Stages**
>
> Yes, we have examined how the model handles repeated object categories that appear in different stages. **As discussed in Section C.3 of the supplementary material**, we selected a model that had been trained across all four stages and visualized its navigation trajectories in tasks from **Stage 1 (toilet) and Stage 2 (bed)**. This analysis helps reveal how well the model retains knowledge of earlier tasks after continual fine-tuning.
> Our TopDown visualizations (Figures 7 and 8) and first-person video comparisons show that the finetuning baseline exhibits clear signs of forgetting, often navigating toward the most recently trained object goal. In contrast, **C-Nav retains prior knowledge and successfully navigates to object goals from earlier stages**, demonstrating greater stability and resistance to forgetting throughout the continual learning process.
>
> To provide quantitative evidence, we further evaluated each model on old tasks across all stages, as shown in the following table:
>
> Performance on Old Tasks at Each Stage (HM3D):
> |Methods|Stage1| |Stage2| |Stage3| |Stage4| |
> |---|---|---|---|---|---|---|---|---|
> | |SR|SPL|SR|SPL|SR|SPL|SR|SPL|
> |Finetuning|69.63|31.25|7.72|2.68|9.83|2.16|9.05|2.55|
> |LoRA|69.63|31.25|17.99|6.27|8.19|2.59|11.70|4.39|
> |Lwf|69.63|31.25|16.19|4.58|8.64|1.74|10.22|2.92|
> |Merge|69.63|31.25|66.03|28.93|41.62|15.40|25.12|7.46|
> |Data Replay|69.63|31.25|54.60|22.57|48.92|19.73|32.94|13.71|
> |C-Nav|69.63|31.25|**61.38**|**26.93**|**52.27**|**22.90**|**42.61**|**17.52**|
>
> These results quantitatively support the superior forgetting resistance of C-Nav compared to all tested baselines.
>
>
> > **Q5: Lack of qualitative analysis and visualization**
>
> As mentioned in Answer1 and Answer2, we have incorporated qualitative visualizations and video comparisons to highlight what the model is learning and how forgetting manifests in practice. Specifically:
> - Figure 7 and Figure 8 in the supplementary materials provide **TopDown trajectory visualizations,** contrasting the finetuning baseline and our C-Nav model.
> - **Four accompanying video clips** show the first-person navigation behavior of both models in various environments, targeting previously seen objects.
> - These comparisons further highlight C-Nav's ability to preserve semantic understanding across tasks, even after multi-stage training.
>
> We hope these additions help clarify our approach and provide a more comprehensive understanding of the model's capabilities
>
> > **Q6: Improve figure explanation**
>
> As also addressed in Answer1 and Answer2, we have **revised Figure 1**, the **Introduction**, and the **case study in the Experimental Results section** to improve clarity and better highlight the challenges of continual object navigation.
>
> > **Q7: Analyze stage-order sensitivity**
>
> We appreciate the reviewer’s insightful suggestion regarding task order sensitivity. To assess the robustness of our method to stage-order permutations, we conducted ablation studies with three different task orderings. The object categories were regrouped across stages.
>
>
> We summarize SR and SPL across all stages in the following table(MP3D):
>
> |Methods|Stage1| |Stage2| |Stage3| |Stage4| |
> |---|---|---|---|---|---|---|---|---|
> | |SR|SPL|SR|SPL|SR|SPL|SR|SPL|
> |Order1|39.88|13.32|37.44|10.02|36.19|9.7|37.22|10.1|
> |Order2|39.88|13.32|35.7|10.51|33.59|9.36|32.12|8.17|
> |Order3|39.88|13.32|39.4|13.32|36.03|8.93|33.17|8.66|
>
> These findings suggest that C-Nav is **robust to different stage sequences**, making it well-suited for real-world continual learning scenarios with dynamic task orders.
>
>
> Thank you again for your valuable suggestions. We will incorporate the relevant additions and improvements in the final version. We hope our response adequately addresses your concerns and would greatly appreciate your consideration for a higher score. Should you have any further questions, we are happy to respond.

---

> > ### Comment · Reviewer_3LhB · 2025-08-09
> > **Post-rrebuttal comment**
> >
> > Thank you for your response, my concerns have been addressed.

---

> ### Author Response · Authors · 2025-08-06
> **Looking forward to your feedback**
>
> Dear reviewer 3LhB:
>
> We sincerely appreciate your insightful and thoughtful comments!  Following your suggestions, we have provided detailed explanations, added visualizations, and included additional experimental results. Specifically, we moved the visualizations from the supplementary materials into the main text, added explanations and intuition for Figure 1, and included results on the impact of learning order as well as a new baseline.
>
> As it is approaching the end of the discussion period (Aug 8 AoE), we sincerely hope you could look through our response and have a further comment at your convenience if you have any questions about the paper. We will do our best to address the issues of the reviewer.
>
>
> Best wishes,
>
> Submission 22093 Authors.

---

> ### Author Response · Authors · 2025-08-08
> **Follow-up on Rebuttal for Submission 22093**
>
> Dear Reviewer 3LhB,
>
> I hope you are doing well.
> As the discussion period will close soon (Aug 8 AoE), I would like to kindly follow up on our rebuttal for Submission 22093.
>
> We have carefully addressed each of your comments with additional explanations, new visualizations, and extended experimental results, as detailed in our response. Your feedback has been invaluable for improving our work, and we would greatly appreciate it if you could take a moment to review our updates and share any further thoughts before the discussion period ends.
>
> Thank you again for your time and constructive feedback.
>
> Best regards,
>
> Submission 22093 Authors.

---

### Official Review · Reviewer_xhkG · 2025-07-03

**Clarity:** 3
**Significance:** 3
**Originality:** 4
**Rating:** 5
**Confidence:** 4

**Summary:**

This paper introduces a new benchmark for studying continual learning in object navigation. As a trial of solving this problem, the authors propose C-Nav, a framework addresses the challenge of catastrophic forgetting in embodied agents by proposing a dual-path anti-forgetting mechanism (feature distillation and feature replay) and an adaptive experience selection strategy. The authors evaluates C-Nav across multiple model architectures, demonstrating superior performance compared to existing continual learning methods.

**Questions:**

Please clarify the weaknesses.

**Ethical Concerns:**

["NO or VERY MINOR ethics concerns only"]

**Final Justification:**

I believe the paper’s most significant contribution is the introduction of a well-designed task setting for evaluating continual-learning capabilities in navigation agents. Prior work has typically assumed that agents are pre-trained once and then deployed directly, with little attention paid to post-deployment adaptation. In practice, continual learning is crucial: real environments often contain situations never seen during pre-training, and adapting to individual user preferences is something pre-training alone cannot achieve. After the authors addressed my main technical concerns in their rebuttal, I would like to recommend acceptance.

**Limitations:**

Yes

**Quality:**

3

**Strengths And Weaknesses:**

# Strengths
1. The paper addresses an underexplored yet critical challenge—continual learning in embodied object navigation—and proposes a novel dual-path framework with strong theoretical grounding and practical relevance.
2. The introduction of a continual object navigation benchmark, built upon HM3D and MP3D datasets, fills an important gap and provides a valuable foundation for future research in this domain.
3. The dual-path anti-forgetting strategy and adaptive experience selection are both innovative and well-motivated, with clear justifications supported by empirical and theoretical insights.
4. Extensive experiments across diverse architectures (RNN, Transformer, BEV, and LLM-based models) and datasets demonstrate the robustness and generalizability of C-Nav, reinforcing its potential for real-world deployment.

# Weaknesses
1. Feature Consistency Loss Concern:
- The justification for the feature consistency loss appears questionable. If the goal is to maintain feature encoding similarity with the original encoder, freezing the encoder would seem like a more straightforward solution. To strengthen the argument, the authors should include an ablation study comparing the proposed approach with a frozen encoder to demonstrate the necessity of loss_KD.
2. Adaptive Experience Feature Selection:
- The proposed adaptive experience feature selection mechanism is innovative and well-designed. However, it would be valuable to compare its performance against simpler alternatives—for example, selecting neighbors based on minimum feature distance and replacing the least relevant samples. Such a comparison would help assess the added benefit of the proposed method's complexity.

---

> ### Author Rebuttal · Authors · 2025-07-29
>
> Thank you for your detailed comments and suggestions on our paper. Your feedback has been invaluable in helping us improve and refine our research. We greatly appreciate your efforts and have provided our responses to each of your points below:
>
>
>
> > **Q1: Feature Consistency Loss Concern**
>
> Thank you for the insightful comment. Our feature consistency loss is designed to learn a shared feature space that remains effective across both past and new tasks. Simply freezing the entire multimodal encoder may preserve old knowledge **but limits the model’s adaptability**, especially when the original feature space does not sufficiently capture new task-specific semantics.
> Instead, we selectively fine-tune the modality-specific projectors within the encoder while applying consistency regularization to softly align the old and new feature spaces. **This design allows the representations to stay connected, yet flexible enough to adapt to new tasks.** It strikes a balance between preserving past knowledge and enabling plasticity for future learning.
> To validate this design choice, we conducted an ablation study on the MP3D Continual ObjectNav dataset by freezing the entire multimodal encoder.
> The results showed a performance drop across learning stages, confirming that full freezing harms generalization and underscoring the necessity of our consistency loss.
>
> | Methods   | Stage1       |        | Stage2       |        | Stage3        |         | Stage4        |         |
> |-----------|--------------|--------|--------------|--------|---------------|---------|---------------|---------|
> |           | SR           | SPL    | SR           | SPL    | SR            | SPL     | SR            | SPL     |
> | Frozen    | 39.88        | 13.32  | 34.64        | 9.72   | 33.35        | 9.44  | 32.8         | 9.32  |
> | Unfrozen  | **39.88**        | **13.32** | **38.88**        | **12.94**  | **38.21**         | **12.35**  | **36.63**         | **11.65**   |
>
>
> > **Q2: Adaptive Experience Feature Selection**
>
> Thank you for your constructive feedback. To validate the effectiveness of our proposed Adaptive Experience Feature Selection mechanism, we conducted an ablation study on the MP3D Continual ObjectNav dataset with two sets of experiments:
>
> * Experiment 1: We compared our method against alternative experience selection strategies, including clustering-based sampling and uniform sampling. For a fair comparison, in the clustering-based approach, we set the number of cluster centers to half the trajectory length and selected frames closest to each cluster center in the feature space as key decision frames.
>
> * Experiment 2: We evaluated the robustness of our method when selecting fewer key frames, by reducing the selection ratio.
>
>
> | Methods   | Stage1       |        | Stage2       |        | Stage3        |         | Stage4        |         |
> |-----------|--------------|--------|--------------|--------|---------------|---------|---------------|---------|
> |           | SR           | SPL    | SR           | SPL | SR           | SPL   | SR            | SPL    |
> |  Clustering-based   | 39.88        | 13.32  | 35.32        | 11.47  | 31.04         | 10.17   | 33.39         | 10.84   |
> | Uniform  | 39.88        | 13.32  | 34.59        | 11.34  | 34.33         | 11.31   | 31.94         | 9.67    |
> | C-Nav     | **39.88**        | **13.32**  | **38.88**        | **12.94**  | **38.21**         | **12.35**   | **36.63**         | **11.65**   |
>
> Results from Experiment 1 demonstrate that our method outperforms the alternatives. From a data distribution perspective, frames that exhibit significant semantic shifts—**such as entering new spaces, approaching target objects, or reaching decision points—are typically rare and can be viewed as outliers in the feature space**. Our method explicitly aims to identify and preserve such outlier frames, which are more informative for navigation decisions. In contrast, **clustering-based or uniform sampling methods tend to select more “normal” frames** that lack critical decision relevance, leading to suboptimal performance.
>
> | Methods   | Stage1       |        | Stage2       |        | Stage3        |         | Stage4        |         |
> |---------|--------------|--------|--------------|--------|---------------|---------|---------------|---------|
> |         | SR           | SPL    | SR           | SPL    | SR            | SPL     | SR            | SPL     |
> | 0.1     | 39.88        | 13.32  | 33.45        | 11.53  | 30.8          | 9.72    | 34.17         | 10.84   |
> | 0.3     | 39.88        | 13.32  | 37.9         | 12.14  | 37.47         | 12.07   | 36.17         | 11.61   |
> | 0.5     | 39.88        | 13.32  | **38.88**        | **12.94**  | **38.21**         | **12.35**  | 36.63         | 11.65   |
> | 1       | 39.88        | 13.32  | 38.37        | 12.29  | 37.32         | 10.97   | **36.81**         | **11.86**   |
>
> Results from Experiment 2 show that our method maintains strong performance even when selecting a smaller proportion of key frames. Notably, **our approach achieves comparable or better results while using only 1/5 of the memory compared to clustering-based or uniform sampling baselines**, highlighting its efficiency and scalability.
>
> Thank you very much for your valuable suggestions, which have been incredibly helpful in improving our work. We will incorporate the above results into the revised version of the paper to further demonstrate the effectiveness of the proposed method. We hope our response addresses your concerns. If you have any additional questions or feedback, we sincerely welcome them and will be glad to provide further clarification. Thank you again for your time and thoughtful review.

---

> > ### Comment · Reviewer_xhkG · 2025-08-08
> >
> > Thank you for your response. My concerns are addressed.

---

### Note · Authors · 2025-08-12

We thank the reviewers and AC for their valuable feedback. C-Nav introduces a continual ObjectNav framework combining dual-path anti-forgetting and adaptive experience selection, enabling agents to adapt to new tasks while retaining past knowledge.
> **New Results**:

(a) Feature Consistency Loss – Freezing the entire multimodal encoder reduces generalization. Our selective fine-tuning with consistency regularization preserves past knowledge and adapts to new tasks.

(b) Adaptive Experience Feature Selection – Our method outperforms clustering-based and uniform sampling.

(c) We conduct ablations on loss weights and retention ratios.

(d) We introduce a comparison with ZSCL, highlighting the unique challenges posed by our embodied setting.

(e) We analyze performance across stages, focusing on how methods handle old versus new tasks.

(f) We explore the impact of task learning order on performance.
> **Visualization Updates**

Trajectory visualizations and videos, moved from supplementary to main text, show that baselines suffer catastrophic forgetting while C-Nav maintains navigation across stages. A new subfigure in the introduction illustrates the continual ObjectNav setup and forgetting challenges.
> **Clarification on Task Setup**

We clarify the distinction between Continual ObjectNav and MultiON: Continual ObjectNav uses multi-phase incremental training, introducing new categories and trajectories over time, with earlier data inaccessible and no persistent maps.
This setup specifically targets the challenges of catastrophic forgetting and the stability–plasticity trade-off, which are less prominent in MultiON.
> **Why Focus on E2E Methods for Continual ObjectNav**

E2E methods remain a key paradigm in ObjectNav, valued for deployment efficiency (faster inference, simpler structure) and strong performance with full data, and are widely studied（UniNavid， TrakeVLA）. However, in Continual ObjectNav they suffer catastrophic forgetting, hindering knowledge retention across stages. C-Nav addresses this challenge, aiming to preserve E2E advantages while enabling long-term retention.

Zero-shot methods avoid forgetting by freezing parameters, but their inability to fine-tune on new data often yields suboptimal performance. The BEV-based E2E outperforms UniGoal by 10% SR on HM3D, with up to 20% gains with more data.

Due to their importance, we further analyze the role of zero-shot methods in Continual ObjectNav in the related work section.

---

### Decision · Program_Chairs · 2025-09-17

**Decision:**

Accept (poster)

**Comment:**

This paper introduces a benchmark for continual learning in object navigation. To make progress on this benchmark, they introduce a new method called C-NAV.

Reviewers saw the strength in this work -- the task paradigm itself is valuable, the proposed method is interesting, and the evaluations are thorough. While there were initially some concerns about the clarity of the presentation and novelty of the method, these were adequately addressed in the rebuttal.